

# A 1-km daily high-accuracy meteorological dataset of air temperature, atmospheric pressure, relative humidity, and sunshine duration across China (1961–2021)

Keke Zhao[1], Denghua Yan[1], Tianling Qin[1], Chenhao Li[1], Dingzhi Peng[2], Yifan Song[1]

[1]State Key Laboratory of Simulation and Regulation of Water Cycle in River Basin, China Institute of Water Resource and Hydropower Research, Beijing 100038, China
[2]College of Water Sciences, Beijing Normal University, Beijing 100875, China

*Corresponding author: Prof. Denghua Yan (yandh@iwhr.com)

**Abstract.** Fine-resolution and high-accuracy meteorological datasets are essential for understanding climate change processes and their cascading impacts on hydrology, water resources management, and ecological systems. In this study, we present a nationwide, high-resolution dataset of six daily meteorological variables across China from 1961 to 2021, including average temperature, maximum temperature, minimum temperature, atmospheric pressure, relative humidity, and sunshine

duration. The dataset was generated through a hierarchical reconstruction framework that utilizes daily observations from 2345 meteorological stations across China, combined with station-level topographic attributes (latitude, longitude, and elevation). By decoding the nonlinear relationships among six meteorological variables and their spatial covariates, the framework enables the generation of gridded daily fields at 1 km resolution with spatial continuity and internal consistency. Validation against 118 in-situ stations confirms that the dataset achieves high accuracy across all variables, with average,

maximum, and minimum temperatures exhibiting minimal errors (median RMSEs: 1.03°C, 1.19°C, 1.34°C; median MEs: -0.09°C, -0.10°C, -0.08°C) and high consistency with in-situ data (median CCs: 1.00, 0.99, 0.99). Atmospheric pressure shows minimal error (median RMSE: 2.48 hPa; median ME: -0.02 hPa) and high consistency (median CC: 0.98). Although relative humidity has slightly weaker accuracy (median RMSE: 6.02%; median ME: -0.5%; median CC: 0.90), it still surpasses standard benchmarks. Sunshine duration maintains high precision (median RMSE: 1.48 h; median ME: 0.05 h;

median CC: 0.93), demonstrating overall excellent product quality. Further comparison reveals that in high-altitude and topographically complex regions, the reconstructed product demonstrates higher actual accuracy than suggested by station-to-grid validation, as spatial mismatches between stations and grid cells lead to systematic underestimation. Free access to the dataset available at https://doi.org/10.11888/Atmos.tpdc.301341 or https://cstr.cn/18406.11.Atmos.tpdc.301341.

## 1 Introduction

With advances in computational power and remote sensing technologies, hydrological modeling has increasingly evolved toward fully distributed simulations, while climate change research continues to expand across broader spatial and temporal





scales. These developments have placed growing demands on the resolution and accuracy of basic meteorological inputs, particularly in ungauged and topographically complex basins such as the Tibetan Plateau. High-resolution and high-quality meteorological datasets are essential for capturing fine-scale climate signals, representing land–atmosphere interactions, and

supporting hydrological, ecological, and environmental assessments.

In recent decades, a wide range of meteorological and environmental variables—such as land and sea surface temperatures, precipitation (King et al., 2003), vegetation indices (Zeng et al., 2022), soil moisture (Brocca et al., 2017), air quality (Martin, 2008), and carbon emissions (Wunch et al., 2017) —have been derived from remote sensing observations and data assimilation systems. These satellite-based products offer broad spatial coverage and long-term continuity, enabling

significant advances in water resources monitoring and drought-related climate assessment, particularly in data-scarce regions (Sheffield et al., 2018). However, despite their strengths, such products often struggle to represent near-surface meteorological conditions with sufficient precision. Their performance is typically constrained by atmospheric interference, cloud contamination, and limited spatial resolution—factors that become particularly problematic in regions with highly variable terrain. As a result, many satellite-derived datasets fail to meet the spatial and temporal requirements of land surface

modeling, hydrological forecasting, and local-scale climate analysis. To mitigate these limitations, assimilation-based approaches have been increasingly adopted to integrate satellite data, reanalysis fields, and ground-based observations for near-surface meteorological forcing generation (Rodell et al., 2004; Laiolo et al., 2015; Liu et al., 2019; Khaki et al., 2020). While these efforts improve data consistency and spatial completeness, significant uncertainties remain—especially in areas like western China, where rugged topography and sparse station distribution pose persistent challenges (Gao and Liu, 2013;

Yang et al., 2013; Wang et al., 2016; Tang et al., 2016; Qi et al., 2018). These limitations underscore the pressing need for regionally tailored, high-resolution meteorological datasets that are capable of capturing local climatic variability and supporting reliable simulation in hydrological modeling, drought risk forecasting, and water resources management.

Recent efforts to generate gridded meteorological forcing datasets in China have primarily followed three methodological approaches. The first approach is based on spatial interpolation of in-situ station data to generate gridded

fields (Li, 2008). However, interpolation methods that do not explicitly account for topographic complexity and environmental gradients often yield limited accuracy, particularly in mountainous regions (Li and Heap, 2011; Yu et al., 2015; Yang and Xing, 2021). To improve spatial realism, elevation-dependent interpolation schemes have been applied to reconstruct precipitation and temperature in regions such as the Heihe River Basin, the Tibetan Plateau, and the headwaters of the Yangtze and Yellow Rivers (Wang et al., 2017; Sun and Su, 2020; Zhao et al., 2022; Zhang et al., 2024). The second

approach involves spatial downscaling and multi-source data fusion. This includes deriving high-resolution fields from coarse-resolution reanalysis or climate datasets, or combining satellite, reanalysis, and station data to reconstruct near-surface meteorological variables. For instance, Li et al. (2014) employed a two-step interpolation method to generate 1 km gridded datasets of air temperature, pressure, humidity, and wind speed across China. Peng et al. (2019) produced monthly gridded temperature and precipitation data for 1901–2017 using delta downscaling applied to CRU and WorldClim inputs.

He et al. (2020) developed the China Meteorological Forcing Dataset (CMFD), which integrates observations from over



1,000 stations with GLDAS and MERRA reanalysis products to provide daily meteorological variables at 0.1° resolution. Zhao et al. (2022) further enhanced precipitation accuracy over the Yarlung Zangbo Basin by correcting and merging multiple satellite precipitation products with in-situ records. The third approach draws upon machine learning techniques to model complex relationships between meteorological variables and spatial covariates. Global satellite-derived precipitation products such as CMORPH (Joyce et al., 2004; Xie et al., 2017) and PERSIANN(Sorooshian et al., 2014; Sadeghi et al., 2019) exemplify early use of neural networks for rainfall estimation. In the Chinese context, recent studies—including those by Wu et al. (2020), Hong et al.(2021), and Jing et al. (2022) —have applied deep learning models to improve the spatial resolution and accuracy of multi-source precipitation datasets. For temperature, Pang et al. (2017) evaluated machine learning methods for downscaling daily mean temperature in the Pearl River Basin using global climate model outputs. Zhang et al. (2021) showed that a gradient boosting approach outperformed traditional reanalysis datasets such as JRA-55 and ERA-Interim over the Tibetan Plateau. He et al. (2022) applied Gaussian process regression to generate the GPRChinaTemp1km dataset, a 1 km resolution monthly temperature product for 1951–2020. However, the development of machine learning-based gridded products for other meteorological variables—such as atmospheric pressure, humidity, sunshine duration, and wind speed—remains limited and warrants further research (Li and Zha, 2018; Liu et al., 2022).

The objective of this study is to develop a high-resolution and accuracy-assessed dataset of daily near-surface meteorological variables across mainland China, suitable for applications in hydrological modeling, environmental monitoring, and climate analysis. To achieve this, a hierarchical and progressive reconstruction framework was implemented to generate gridded fields of six variables—average, maximum, and minimum air temperature, atmospheric pressure, relative humidity, and sunshine duration. These variables were reconstructed at approximately 2 meters above ground level at a spatial resolution of 1 km, and the framework allows for adaptation to other spatial scales supported by the digital elevation model (DEM). A multilayer perceptron (MLP) regression model was used to capture nonlinear relationships between station observations and topographic predictors (e.g., latitude, longitude, and elevation), enabling fine-scale reconstruction across complex terrain.

## 2 Materials

### 2.1 Training and validation data from CMA

Daily records of meteorological variables—including longitude, latitude, elevation, average temperature, maximum temperature, minimum temperature, atmospheric pressure, relative humidity, and sunshine duration—were obtained from 2,440 meteorological stations operated by the China Meteorological Administration (CMA) for the period 1961–2021. To support independent model validation, a total of 95 stations were selected as evaluation sites based on three principles: (1) ensuring geographical representativeness in terms of longitude, latitude, and elevation; (2) in densely monitored areas such as eastern China, a greater number of evaluation stations were retained without significantly reducing the size of the training

dataset; and (3) in sparsely monitored regions such as western China (including Tibet and Xinjiang), fewer stations were assigned to the evaluation set in order to preserve sufficient data for model training. The remaining 2,345 stations were used exclusively for training purposes. The spatial distribution of both training and evaluation stations is shown in Figure 1.

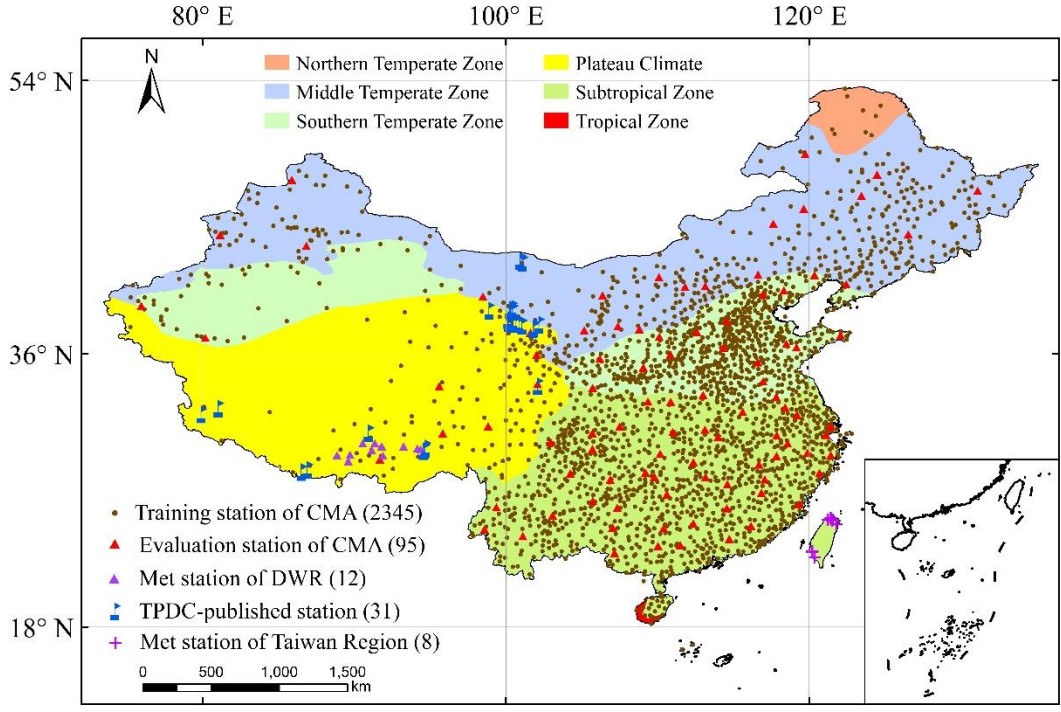


**Figure 1: The spatial distribution of training and evaluation meteorological stations in China.**

For the years 2020 and 2021, daily records are limited to air temperature, as measurements of atmospheric pressure, relative humidity, and sunshine duration are unavailable during this period. Due to variations in the temporal coverage of individual stations, the amount of available daily data for model training and evaluation also differs across sites. The

temporal distribution of operational meteorological stations from 1961 to 2021 is presented in Figure 2.

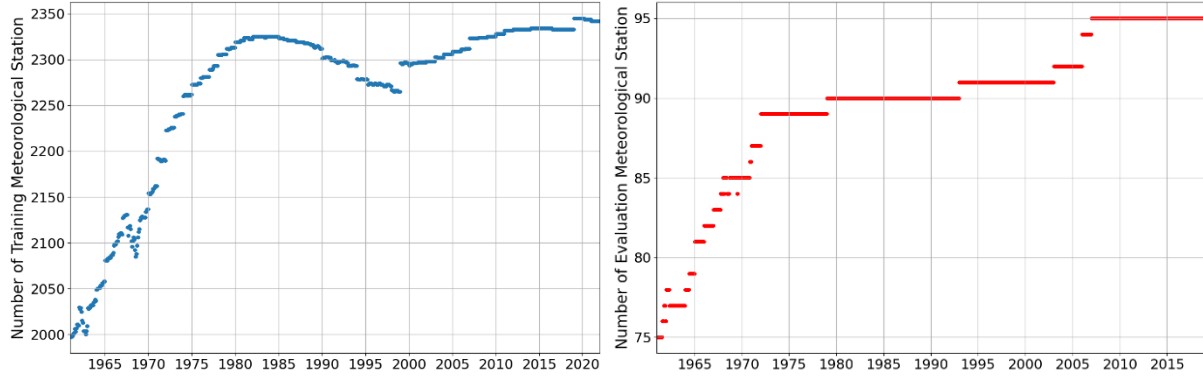

**Figure 2: The spatial distribution of training and evaluation meteorological stations in China.**





## 2.2 Validation data from supplementary ground-based observations

### 2.2.1 Ground observations provided by DWR

To address the limited spatial coverage of validation stations in the Tibet region, daily average temperature observations from 12 ground-based meteorological stations were obtained from the Department of Water Resources (DWR). These supplementary data enhance the robustness of model evaluation in western China. The locations of the DWR stations are shown in Figure 1, and metadata for each station are provided in Table 1.

**Table 1: Detailed information on records from DWR ground-based meteorological observation stations.**

| Number | Station Name | Station Type | Time Range | Element Type |
|---|---|---|---|---|
| 1 | Nugesha | Meteorological | 2001.1.1~2003.12.31 | |
| 2 | Yangcun | Meteorological | 2001.1.1~2003.12.31 | |
| 3 | Nuxia | Meteorological | 2001.1.1~2003.12.31 | |
| 4 | Jiangzi | Meteorological | 2001.1.1~2001.12.31 | |
| 5 | Rikaze | Meteorological | 2001.1.1~2001.12.31 | |
| 6 | Pangduo | Meteorological | 2001.1.1~2003.12.31 | Average temperature |
| 7 | Tangjia | Meteorological | 2001.1.1~2003.12.31 | |
| 8 | Lhasa | Meteorological | 2001.1.1~2003.12.31 | |
| 9 | Yangbajing | Meteorological | 2001.1.1~2003.12.31 | |
| 10 | Gongbujiangda | Meteorological | 2001.1.1~2003.12.31 | |
| 11 | Gengzhang | Meteorological | 2001.1.1~2003.12.31 | |
| 12 | Bayi | Water Level | 2001.1.1~2003.5.31 | |

### 115 2.2.2 Literature-based datasets from the National Tibetan Plateau Data Center

To supplement observational data for the evaluation of gridded meteorological products, a variety of station-based datasets were obtained from the National Tibetan Plateau Data Center (TPDC, http://data.tpdc.ac.cn), as represented by the blue flag symbols in Figure 1. These include: (1) a publicly available dataset of hourly land–atmosphere interaction observations from 12 field stations (Ma et al., 2024) , covering the period 2005–2021, from which 2 stations were selected after quality control

for use as independent validation sites; (2) data from 18 stations within the HiWATER hydrometeorological observation network in the upper reaches of the Heihe River Basin (Liu et al., 2018; Che et al., 2019); and (3) additional station-based records from 11 individual stations, including 2 stations from Zhang (2018a, 2018b); 3 stations from Gao (2018); 2 stations from Luo (2019);  and 1 station each from Ma (2018), Wang and Wu (2019), Luo and Zhu (2020), and Meng and Li (2023).

### 2.2.3 Validation data from GSOD

The Global Surface Summary of Day (GSOD) dataset, compiled by the National Centers for Environmental Information (NCEI), is based on international data exchanges conducted under the World Meteorological Organization (WMO) World



Weather Watch Program. This dataset provides daily summaries of 18 surface meteorological variables from more than 9,000 global stations, with records available from 1929 to the present. Observation data from eight meteorological stations in the Taiwan region were obtained from the NCEI online archive (https://www.ncei.noaa.gov/access/search/data-

search/global-summary-of-the-day) and processed for use in validation. Detailed metadata and data availability for these stations are summarized in Table 2.

**Table 2: Detailed meteorological data from 8 meteorological stations of Taiwan Region.**

| Number | Station name | Time range | Element Type |
|--------|--------------|------------|--------------|
| 1 | TAINAN 593580 | 1973.1.7~1998.12.31 | |
| 2 | SUNGSHAN | 1961.1.1~2021.12.31 | |
| 3 | TANSHUI | 1973.1.7~1977.10.31 | |
| 4 | ILAN CITY | 1973.1.7~1998.12.31 | Average temperature, maximum temperature, minimum temperature |
| 5 | TAIBEI | 1973.1.1~1998.12.31 | |
| 6 | TAINAN | 1961.1.1~2021.12.31 | |
| 7 | TAOYUAN | 1961.1.1~1999.7.26 | |
| 8 | KAOHSIUNG INTERNATIONAL | 1973.1.1~2021.12.31 | |

**2.3 Static geospatial input: SRTM DEM (1km)**

The Digital Elevation Model (DEM) provides high-resolution geographic information—including longitude, latitude,

and elevation—that is required for the spatial reconstruction of meteorological variables. In this study, the DEM was used as an essential input for the reconstruction model to ensure spatial consistency and accuracy. Although the model supports flexible output resolutions, a spatial resolution of 1 km was selected to balance computational efficiency and data detail. The DEM used herein was derived by resampling the latest version of the Shuttle Radar Topography Mission (SRTM) data (version 4.1), as provided by the Consortium for Spatial Information of the CGIAR (Jarvis et al., 2008).

**2.4 Climate regionalization map of China**

The Climate Regionalization Map of China, compiled by the China Meteorological Administration in 1978 using climate data from 1951 to 1970, divides the country into nine climatic zones. The dataset is publicly available via the Resource and Environmental Science Data Platform (https://www.resdc.cn/). For the purpose of comparative analysis of regional climatic patterns, the four subtropical zones—Northern Subtropical, Middle Subtropical, Southern Subtropical, and Southern

Subtropical—were merged into a single Subtropical Zone. The revised classification scheme consists of six zones: Plateau Climate Zone, Northern Temperate Zone, Middle Temperate Zone, Southern Temperate Zone, Subtropical Zone, and Middle Tropical Zone, as illustrated in Figure 1.



## 2.5 Existing gridded products for comparison

To assess the reliability and application potential of the reconstructed meteorological variables, representative and widely
used gridded datasets were selected for comparison based on their scientific relevance and availability. Specifically, for average temperature, atmospheric pressure, and relative humidity, we employed the latest version of the China Meteorological Forcing Dataset (CMFD 2.0), whose earlier versions have been extensively used in land surface, hydrological, and ecological modeling over China (He et al., 2020). As CMFD 2.0 currently does not include sunshine duration, the Homogenized Daily Sunshine Duration (SSD) dataset developed by He et al. (2025) was additionally
incorporated to enable comparative evaluation of the spatial and temporal consistency of sunshine-related variables. Given the scarcity of publicly available sunshine duration datasets, the SSD provides a valuable reference for national-scale assessments over China.

The CMFD 2.0 (He et al., 2024) provides high-resolution (0.1°), 3-hourly gridded meteorological data for the period 1951–2020, covering the land area between 70°E–140°E and 15°N–55°N. It includes near-surface temperature, surface
pressure, specific humidity, wind speed, radiation, and precipitation. Compared to previous versions, CMFD 2.0 incorporates ERA5 reanalysis and station observations through updated data sources and artificial intelligence techniques, particularly for radiation and precipitation variables. It also introduces metadata on station relocations and expands the spatial coverage beyond China's borders, thereby improving temporal consistency and cross-regional applicability.

The SSD dataset (He, 2024) offers spatially continuous and temporally consistent records of daily sunshine duration
across China from 1961 to 2022, at a spatial resolution of 2.0° × 2.0°. It was developed using observations from more than 2,200 meteorological stations and addresses inhomogeneities in the raw records caused by non-climatic factors, including nationwide station relocations and the widespread transition from manual to automatic instruments in 2019.

## 3 Methodology

### 3.1 MLP-based hierarchical progressive reconstruction framework

The reconstruction of near-surface meteorological fields in this study is based on multilayer perceptron (MLP) models—a class of deep feedforward neural networks capable of capturing complex nonlinear relationships through layered transformations (Bisong, 2019). Each MLP consists of an input layer, multiple hidden layers, and an output layer, and is trained using a two-phase process: feedforward propagation, in which input data are transmitted through the network to produce predictions, and backpropagation, during which model parameters are iteratively adjusted to minimize prediction
errors. This learning mechanism enables MLPs to extract spatial and statistical patterns from high-dimensional data while maintaining strong generalization capability. Owing to these characteristics, MLPs have been successfully applied in diverse domains such as medical diagnostics (Karayilan and Kilic, 2017; Desai and Shah, 2021), finance (Duan, 2019; Weytjens et al., 2021), and hydrology (Singh et al., 2012; Choubin et al., 2016; Ren et al., 2020).



In this study, MLP models serve as the computational foundation of the hierarchical progressive reconstruction framework developed to generate high-resolution, spatially complete datasets of near-surface meteorological variables. This framework is designed to address both variable interdependence and geographic heterogeneity by reconstructing each target variable sequentially using a tailored set of spatial and meteorological predictors. As illustrated in Figure 3, it consists of two functional modules: a training module and a reconstruction module. The training module learns nonlinear spatial mapping functions from in-situ station data, capturing daily spatial patterns across complex terrain. The reconstruction module then applies the trained parameters to gridded predictor layers to generate continuous spatial fields at the desired resolution. To ensure both the accuracy and feasibility of the reconstruction, input features are selected based on their relevance to the spatial distribution of each variable and the availability of high-resolution gridded data. Topographic predictors (latitude, longitude, and elevation) are used consistently throughout the framework, while previously reconstructed meteorological variables are incorporated as auxiliary inputs in subsequent steps.

The hierarchical reconstruction framework comprises four sequential steps, each targeting a specific meteorological variable—(a) air temperature, (b) atmospheric pressure, (c) relative humidity, and (d) sunshine duration. This ordering is guided by both physical dependencies and statistical considerations, allowing upstream variables to serve as essential inputs for reconstructing downstream variables. In the first step, air temperature is reconstructed using only geographic predictors—longitude, latitude, and elevation. Although solar radiation and land surface characteristics, which fundamentally shape temperature patterns, are not explicitly included (Peixoto and Oort, 1992; Hartmann, 2016), these geographic features serve as effective proxies for capturing dominant spatial gradients. In the second step, atmospheric pressure is modeled using a three-layer MLP, incorporating geographic variables and temperature. Atmospheric pressure is jointly determined by air density and gravitational acceleration, both of which vary with temperature and elevation due to their effects on the atmospheric hydrostatic balance (Mason et al., 2016). Including temperature as a predictor thus improves the model's ability to reproduce its spatial variability. The third step addresses relative humidity, modeled using a four-layer MLP with geographic predictors, temperature, and atmospheric pressure as inputs. Relative humidity depends on both actual and saturation vapor pressures (Wallace and Hobbs, 2006; Mason et al., 2016) ; the former is partially influenced by atmospheric pressure, while the latter is primarily governed by temperature and increases exponentially according to the Clausius–Clapeyron relationship. Incorporating both temperature and pressure enhances the model's ability to capture the complex spatial behavior of humidity. Building on the preceding steps, the final reconstruction targets sunshine duration, which is influenced by the combined effects of the solar astronomical position, atmospheric radiative processes, and synoptic-scale weather systems. According to WMO (2023), sunshine duration is defined as the total time during which direct solar irradiance exceeds 120 W/m². Geographic predictors provide the spatial context, while temperature, pressure, and humidity reflect dynamic atmospheric states and cloud-related feedbacks. These variables are physically grounded and observationally accessible. A four-layer MLP model is therefore employed in the final step to reconstruct the spatial distribution of sunshine duration.



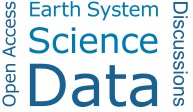

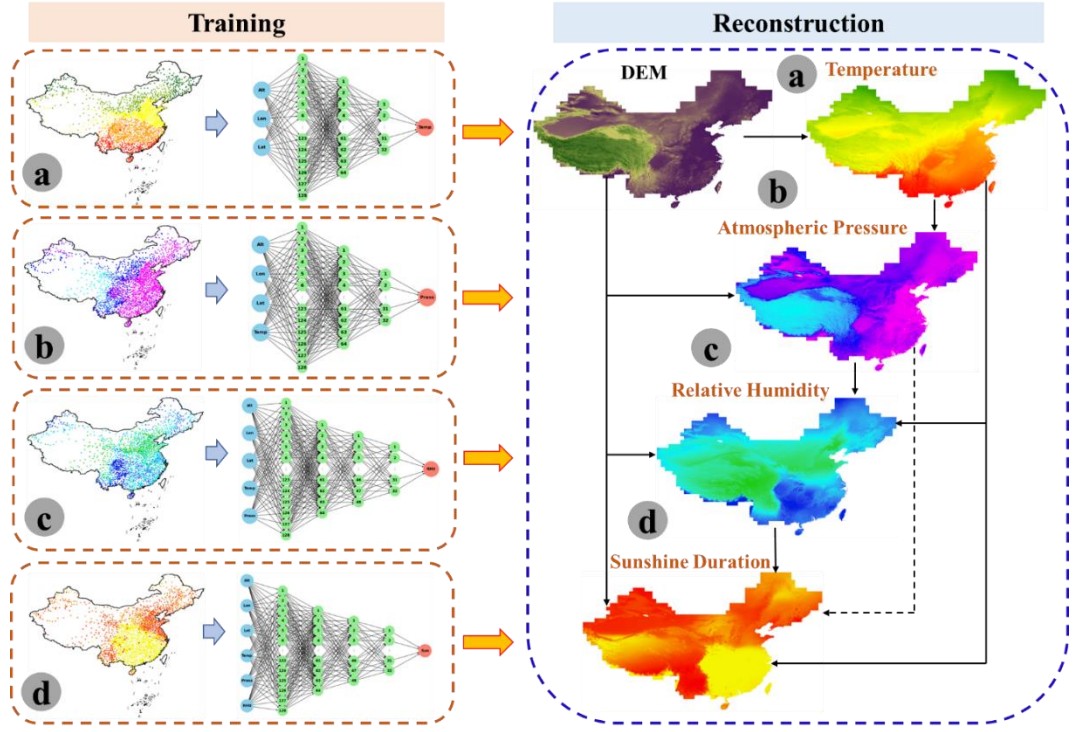

**Figure 3: MLP-based hierarchical progressive reconstruction framework for China.**

Overall, this progressive framework ensures that each reconstruction step is informed by physically meaningful and context-specific predictors. By integrating the hierarchical dependencies among meteorological variables, the approach yields spatially complete and physically consistent gridded datasets suitable for large-scale climate and environmental applications.

## 3.2 Evaluation metrics

In this study, four evaluation metrics were employed: Mean Error (ME), Mean Squared Error (MSE), Root Mean Square Error (RMSE), and Correlation Coefficient (CC). These metrics were utilized in two distinct phases: the MLP model training phase and the meteorological products evaluation phase. The formulas for the four metrics are as follows:

$$ME = \frac{1}{n}\sum_{t=1}^{n}\left(Y_t - \widehat{Y}_t\right)$$

$$MSE = \frac{1}{n}\sum_{t=1}^{n}\left(Y_t - \widehat{Y}_t\right)^2$$

$$RMSE = \sqrt{MSE}$$

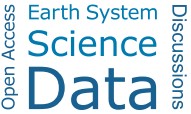

$$CC = \frac{\sum_{t=1}^{n}(Y_t - \overline{Y})(\widehat{Y}_t - \overline{\widehat{Y}})}{\sqrt{\sum_{t=1}^{n}(Y_t - \overline{Y})^2}\sqrt{\sum_{t=1}^{n}(\widehat{Y}_t - \overline{\widehat{Y}})^2}}$$

Where $n$ is the total number of days in the time series; $t$ stands for the $t$-th day; $Y_t$ and $\overline{Y}$ stand for the in-situ value of the target variable and the mean in-situ value of the target variable, respectively; and $\widehat{Y}_t$ and $\overline{\widehat{Y}}$ stand for the model's estimated value and the mean estimated value, respectively.

During the training phase, MSE was used as the loss function to measure and optimize the performance of the MLP
model. Upon completion of the training, ME and CC were computed between the estimated outputs—derived from the model parameters at the optimal training state— and in-situ records of the target variable, with particular emphasis on CC to ensure comprehensive model performance evaluation. If the MSE was low but the CC was poor, the hyperparameters of the deep learning model were adjusted, and training continued until satisfactory results were achieved.

In the subsequent evaluation phase of the meteorological reconstruction products, RMSE, ME, and CC were calculated
between in-situ records and corresponding grid estimates. These metrics effectively validated the accuracy and reliability of the reconstruction products, confirming discrepancies with the observed data.

## 4 Results and discussion

### 4.1 MLP training and test results

To evaluate the generalization capability of the reconstruction models and prevent overfitting, we randomly assigned 10% of
the daily in-situ observations from 1961 to 2021 to the test dataset using a fixed random seed, with the remaining 90% used for training. Figure 4 presents the performance metrics of the daily MLP models across six meteorological variables: average temperature, maximum temperature, minimum temperature, atmospheric pressure, relative humidity, and sunshine duration. Three standard evaluation metrics are used: ME (Figure 4(a)), MSE (Figure 4(b)), and CC (Figure 4(c)). The mean values of all metrics are highly consistent between training and test phases, indicating strong generalization and no evidence of
overfitting. These results confirm the stability and precision of the deep learning-based hierarchical progressive reconstruction framework. Notable deviations across all metrics are limited to a very small number of days and are primarily attributed to substantial gaps in the in-situ observations.





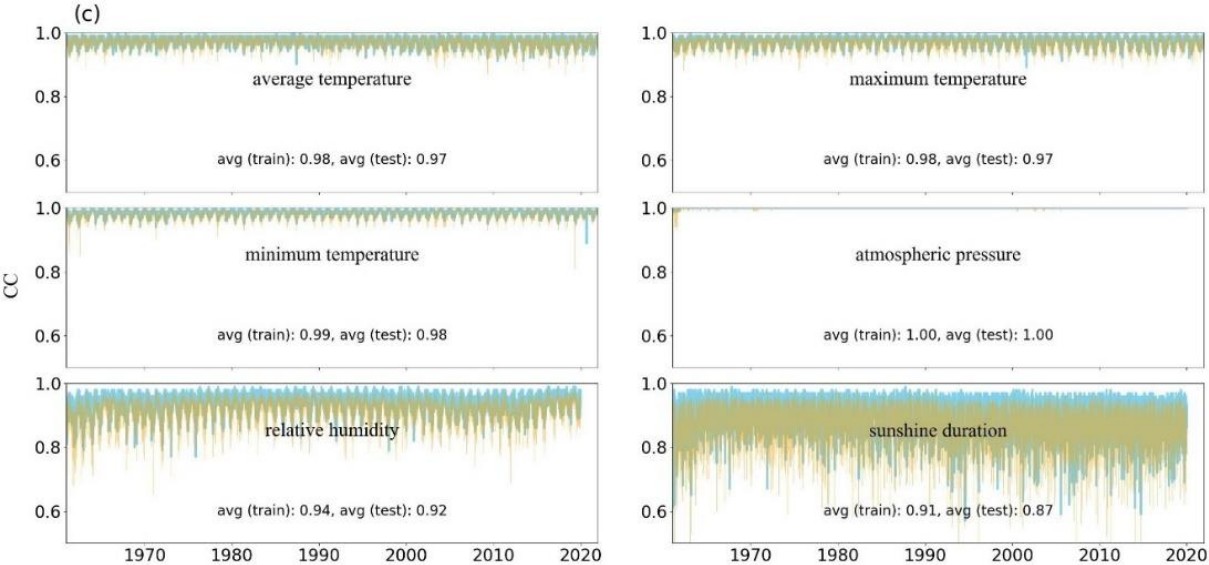


**Figure 4: Line graphs of metrics (MSE, ME, CC) for optimal parameters in daily training and testing of MLP models from 1961 to 2021.**

The ME values are close to zero for all variables in both phases. Specifically, the mean ME for maximum and minimum temperatures is exactly 0 °C, while the other four variables also show near-zero mean errors, with at least one phase yielding

a mean ME of 0. The range of ME values is also narrow. During training, ME ranges from –0.49 °C to 0.46 °C for average temperature, –3.55 hPa to 2.61 hPa for atmospheric pressure, –2.15% to 1.96% for relative humidity, and –0.54 h to 0.50 h for sunshine duration. The test phase exhibits even narrower ME ranges: –0.32 °C to 0.36 °C (average temperature), –2.25 hPa to 1.94 hPa (atmospheric pressure), –1.83% to 1.49% (relative humidity), and –0.42 h to 0.41 h (sunshine duration). These results suggest minimal systematic bias in the model predictions across all variables. The MSE, which emphasizes the

impact of large residuals by squaring the error magnitude, consistently exceeds the ME across all variables. As shown in Figure 4(b), the daily MSE values are low in both phases, with only a slight increase in the test phase. Temperature-related variables—including average, maximum, and minimum temperature—exhibit low and stable MSE values, with means below 1 °C² and only minor differences (typically 0.1 °C²–0.3 °C²) between training and test phases. This indicates that the model captures temperature dynamics with high accuracy and strong generalization. For atmospheric pressure, which inherently exhibits a larger numerical scale, the mean MSE values remain relatively low—6.9 hPa² in the training phase and 8.5 hPa² in

the test phase. Notably elevated MSE values are observed only on a few days in 1961, primarily due to substantial gaps in the observed atmospheric pressure records. Relative humidity and sunshine duration also show consistently low error levels, with training phase MSEs of 14.1 %² and 1.2 h², and slightly higher values of 20.7 %² and 1.8 h² in testing phase. Analysis of the CC value indicates strong agreement between model estimates and observed values across all variables. Notably, atmospheric pressure achieves perfect agreement, with a mean CC of 1.00 in both phases. Average, maximum, and minimum

temperatures also show consistently high correlations, with mean CCs of 0.98, 0.98, and 0.99 in the training phase, and 0.97,



0.97, and 0.98 in the testing phase. Although the CCs for relative humidity and sunshine duration are slightly lower, they remain strong—0.94 and 0.91 in training, and 0.92 and 0.87 in testing, respectively.

Collectively, the results highlight the proposed framework's ability to accurately identify and reconstruct the spatial
structures of diverse meteorological variables, demonstrating strong generalization across different element types and conditions.

## 4.2 Validation of gridded meteorological element products using in-situ data

An independent validation was conducted using long-term in-situ records from 118 meteorological stations, as described in Sections 2.1 and 2.2. These stations were entirely excluded from the model training and testing phases, and their
observations served as reference data for an objective evaluation of the reconstructed products' accuracy and spatial generalizability. The validation results confirm that the reconstructed meteorological products achieve high overall accuracy, with particularly strong performance in regions with dense training data. Notably, even in areas with sparse or absent observations—such as northwestern China and Taiwan—the model maintains stable and reliable performance, indicating strong spatial generalizability and a capacity to extrapolate beyond the training domain. This highlights the potential of the
proposed framework for broad application in diverse climatic and geographic settings. Model performance was quantified by calculating RMSE, ME, and CC between the 1 km gridded estimates and the corresponding station observations. The evaluation metrics were visualized through box plots (Figure 5) and spatial distribution maps (Figures 6).

As shown in the box plots of RMSE, ME, and CC (Figure 5), the reconstructed products for average, maximum, and minimum air temperature exhibit minimal errors and excellent consistency with in-situ observations. Median RMSEs are
1.03°C, 1.19°C, and 1.34°C, respectively; median MEs are close to zero (−0.09°C, −0.10°C, and −0.08°C); and median CCs are exceptionally high (1.00, 0.99, and 0.99). Despite its inherently larger magnitude, atmospheric pressure also demonstrates high precision, with a median RMSE of 2.48 hPa, ME of −0.02 hPa, and CC of 0.98. In comparison, the relative humidity product shows moderately lower agreement with observations, reflected in a median RMSE of 6.02%, ME of −0.5%, and CC of 0.90. However, since it is primarily used as an input for the reconstruction of sunshine duration, its
effect on overall model performance is limited. Indeed, the sunshine duration product demonstrates higher accuracy, with a median RMSE of 1.48 h, ME of 0.05 h, and CC of 0.93. Although relative humidity exhibits slightly weaker performance than other variables, its accuracy still exceeds typical benchmarks and remains suitable for practical applications.

The spatial distribution of RMSE, ME, and CC for all six meteorological variables is further illustrated in Figures 6. Consistent with expectations, the Subtropical and Southern Temperate Zones in southeastern China (STZ-southeastern China)
display the best performance across all variables, largely due to the high density of training stations in these regions. In contrast, performance metrics are relatively lower in the Middle Temperate, Southern Temperate, and Plateau Climate Zones of northwestern China (MSPZ–northwest China), as well as in Taiwan, where no stations were included in training. Nevertheless, model performance in these regions remains robust. Notably, despite the absence of training data in Taiwan, the MLP model accurately reconstructs air temperature in that region, suggesting strong spatial generalizability.





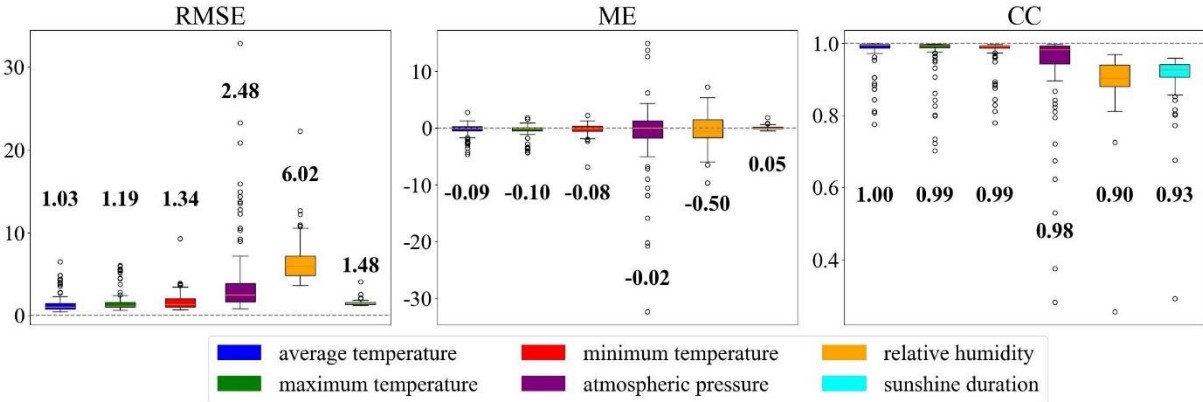


**Figure 5: Box plots of RMSE, ME, and CC for grid-modelled data of six meteorological element products and in-situ data.**

For temperature variables, both Figure 5 and Figure 6 indicate minimal spatial variation, with most RMSEs, MEs, and CCs in STZ southeastern China and MSPZ northwest China falling within the ranges of 0.49 °C to 2 °C, −2 °C to 2 °C, and 0.95 to 1.00, respectively. A few outliers, primarily located in the Tibetan Plateau, Xinjiang, and Taiwan, fall outside these

ranges. Specifically, temperature errors in Taiwan range from 3.3 °C to 6 °C for RMSE, −0.5 °C to −4 °C for ME, and 0.7 to 0.9 for CC, indicating a general underestimation of air temperature in this region. For the atmospheric pressure product, RMSE, ME, and CC values in STZ southeastern China generally range from 0.8 hPa to 15 hPa, −5 hPa to 5 hPa, and 0.85 to 1.00, respectively. In MSPZ northwest China, most ME values range from −32 hPa to 0 hPa, indicating a slight tendency toward underestimation. For the relative humidity product, RMSE values indicate relatively larger errors in MSPZ northwest

China, generally ranging from 8% to 12%, with a clear tendency toward underestimation, as most ME values fall between −9.5% and 0%. This underestimation trend is less evident in the Tibetan Plateau. In the eastern half of China, errors are smaller, with RMSE values typically ranging from 3.6% to 8% and ME values between −3% and 4%. No distinct spatial pattern of underestimation or overestimation is observed. Similarly, CC values show no clear spatial variability across the country, mostly ranging from 0.80 to 1.00, with only two isolated stations exhibiting lower values. For the sunshine duration

product, RMSE, ME, and CC values exhibit minimal spatial variability across China. RMSE values generally range from 1.2 h to 2.0 h, ME values from −0.4h to 0.5h, and CC values from 0.80 to 1.00. Values beyond these ranges are observed only at a few isolated stations.







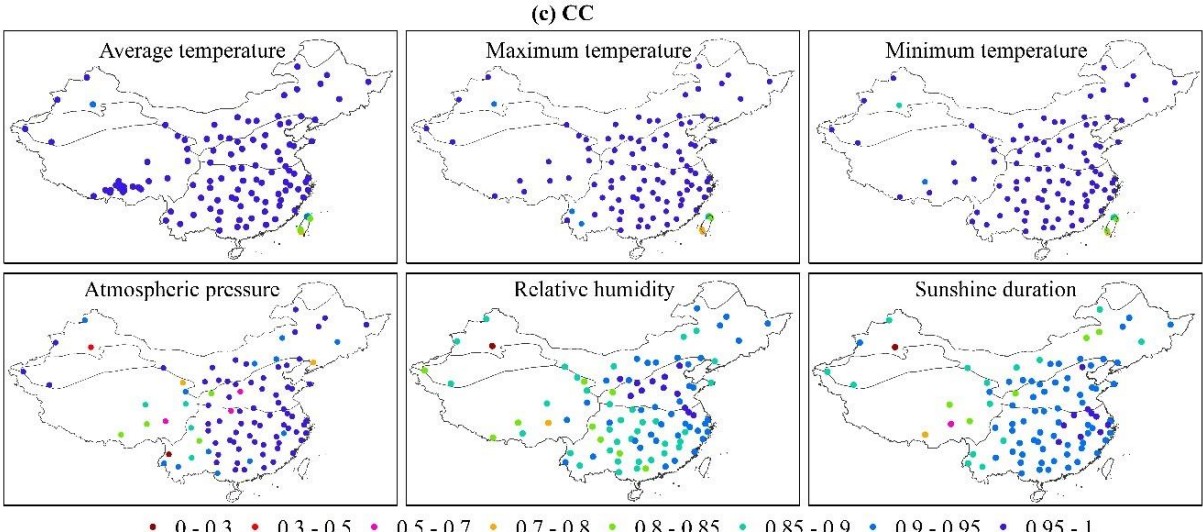

**Figure 6: Distribution maps of RMSE (a), ME (b) and CC (c) between grid-modelled data of six meteorological element products and in-situ data.**

## 4.3 Evaluation and comparison against existing gridded products

### 4.3.1 Average temperature, atmospheric pressure, relative humidity

Although 95 CMA stations were initially reserved for validating the gridded meteorological products developed in this study, they were not used in the evaluation of CMFD 2.0 due to the lack of publicly available information on the station sources used in its construction. This raised concerns that some or all of these CMA stations might have already contributed to the CMFD 2.0. To avoid potential data overlap and ensure an objective and independent evaluation, these stations were excluded from the validation analysis. Instead, observational data from 51 ground stations introduced in Sections 2.2.1, 2.2.2, and 2.2.3 were used to assess the accuracy of the reconstructed meteorological variables against CMFD 2.0. These stations provided daily records for one to three of the following variables: average temperature (48 stations), atmospheric pressure (25 stations), and relative humidity (29 stations). As maximum/minimum temperature and sunshine duration were largely unavailable at these sites and not included in CMFD 2.0, the evaluation focused exclusively on the three core variables.

As shown in Figure 7, except for atmospheric pressure—where CMFD 2.0 exhibits a higher median CC value (0.96) than this reconstructed dataset (0.87)—the gridded meteorological dataset developed in this study demonstrates generally comparable or slightly improved performance relative to CMFD 2.0 in terms of median RMSE, ME, and correlation coefficient across the evaluated variables. Notably, although the correlation for atmospheric pressure is marginally lower in the dataset developed in this study, it yields substantially smaller errors, with median RMSE and ME of 3.61 hPa and –0.61 hPa for this dataset, and 17.14 hPa and 9.41 hPa for CMFD 2.0, respectively. For average temperature and relative humidity, the two gridded products exhibit similar median CC values. However, the reconstructed dataset yields consistently lower median RMSE and ME, suggesting slightly improved accuracy. Specifically, the values for temperature are 1.98 °C and –





0.21 °C, compared to 2.08 °C and –0.46 °C for CMFD 2.0. For relative humidity, the corresponding values are 10.75 % and –1.05 % for the reconstructed dataset, while CMFD 2.0 reports 11.12 % and –2.40 %.

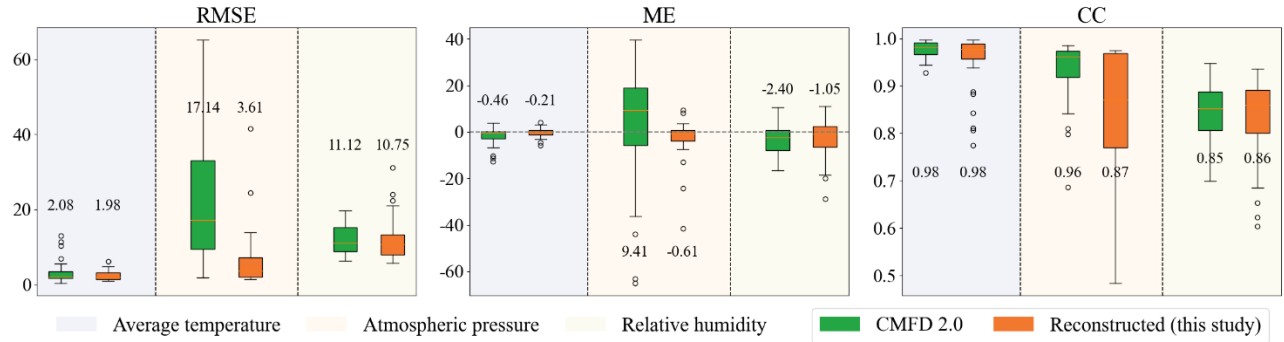

**Figure 7: Boxplot comparison of RMSE, ME, and CC for average temperature, atmospheric pressure, and relative humidity between CMFD 2.0 and the reconstructed dataset developed in this study.**

These findings are particularly evident in high-altitude regions represented by 51 validation sites predominantly located in the southern Tibetan Plateau and the Heihe River Basin, where the gridded fields of average temperature, atmospheric pressure, and relative humidity developed in this study demonstrate good agreement with station observations. Compared with CMFD 2.0, a widely used multi-source reanalysis product in China, the reconstructed dataset provides improved spatial resolution and slightly enhanced accuracy at these alpine sites. These results suggest the potential of the dataset to support regional-scale hydrometeorological studies in cold and topographically complex environments.

**4.3.2 Sunshine duration**

Figure 8 presents a comparison of sunshine duration estimates between the SSD product and the reconstructed dataset developed in this study, evaluated using 95 validation stations. The median values of RMSE (1.48 h) and CC (0.93) are identical for both datasets, while the ME shows only a slight difference (0.02 h for SSD and 0.05 h for the reconstructed dataset), indicating a comparable bias level. The boxplots reveal subtle differences in distribution characteristics: the reconstructed dataset shows slightly narrower interquartile ranges, while the SSD product exhibits fewer outliers in RMSE and CC. Overall, the reconstructed sunshine duration dataset yields accuracy comparable to that of the SSD product, while providing notably higher spatial resolution (1 km).

A comparative accuracy assessment of the reconstructed sunshine duration dataset was conducted using observations from 95 CMA stations, with the SSD product serving as the benchmark. Notably, some of these validation stations may have been assimilated during the development of the SSD product, whereas the reconstructed dataset presented in this study was developed without incorporating any of these stations in its training phase, thereby ensuring a higher degree of independence in validation. Although fully independent observations remain limited, the two datasets exhibit strong agreement across RMSE, ME, and CC metrics. This consistency underscores the robustness and generalization capability of the proposed reconstruction method. Furthermore, it demonstrates that, even under partially non-independent validation conditions, the



reconstructed sunshine duration fields attain an accuracy level comparable to that of the benchmark product published in *Earth System Science Data*, supporting their utility for mesoscale to fine-scale hydrometeorological applications in topographically complex regions.

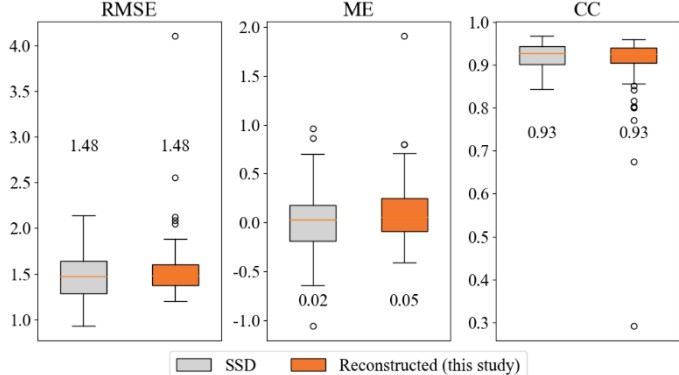

**Figure 8: Boxplot comparison of RMSE, ME, and CC for sunshine duration between SSD and the reconstructed dataset developed in this study.**

### 4.4 Influence of elevation mismatch on validation accuracy

In certain areas of the MSPZ northwest China region—particularly in Tibet and Xinjiang—the validation metrics presented in Section 4.2 indicate relatively lower performance. To examine whether this discrepancy is related to spatial inconsistencies between meteorological station elevations and those of the corresponding grid cells, we analyzed elevation differences using the 1 km DEM. Specifically, elevation mismatch was calculated as the difference between the recorded elevation of the 118 validation stations and the DEM-derived elevation of their corresponding grid cells, as shown in Figure 9. A total of 28 stations were identified where the elevation difference exceeded 50 m, marked with red numbered symbols in Figure 9(a). These stations are primarily located in high-relief regions, and while not all lie within the Plateau Climate Zone, that zone exhibits the largest elevation mismatches. Figure 9(b) ranks these stations by descending elevation difference, with the maximum discrepancy of 591m observed at Station 1 (DWR: Pangduo), followed by 323m at Station 2 (CMA: Tianshan Daxigou) in Xinjiang. To assess the influence of elevation mismatch on validation accuracy, we used the actual longitude, latitude, and elevation of the 28 stations as inputs to the reconstruction module of the MLP-based framework. For each station, the long-term time series of six meteorological variables—average temperature, maximum temperature, minimum temperature, atmospheric pressure, relative humidity, and sunshine duration—were estimated. RMSE, ME, and CC values were then calculated by comparing these station-based estimates with the corresponding in-situ observations, and further compared with the original grid-based validation results.





395

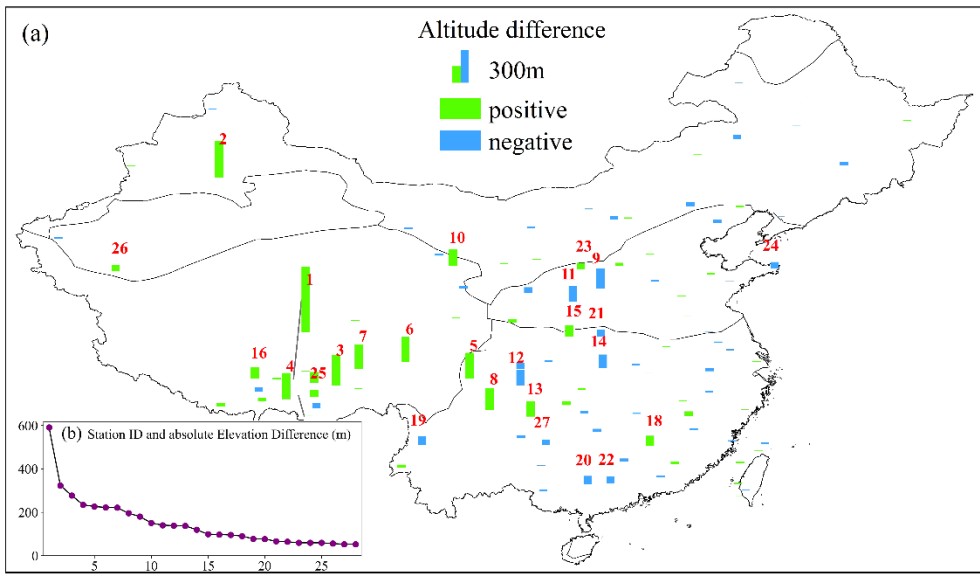

**Figure 9: Distribution of elevation differences between station elevations and corresponding DEM grid values (Red numbers mark elevation differences greater than 50m).**

Figure 10 summarizes the key findings. First, for average temperature, maximum temperature, minimum temperature, and atmospheric pressure, the RMSE and ME between in-situ observations and station-based estimates show substantially

400 greater improvement than those derived from gridded estimates. Notably, the magnitude of improvement increases with larger absolute elevation differences. While relative humidity and sunshine duration also exhibit improvements, the extent is considerably smaller. In contrast, the CCs show modest increases across variables, though the improvement is less pronounced than that observed in error metrics. These results confirm that the MLP-based reconstruction framework yields more accurate estimates than the grid-based approach discussed in Section 4.2, particularly in high-altitude and

405 topographically complex regions.

These findings also highlight potential limitations in using in-situ station data to validate gridded meteorological products—especially in regions with coarse spatial resolution or substantial terrain variability. As grid size increases, spatial mismatches between stations and grid cell averages (in terms of latitude, longitude, and elevation) become more pronounced. Even at 1 km resolution, notable elevation mismatches were observed in high-altitude areas. For variables highly sensitive to

410 elevation and geographic location—such as air temperature and atmospheric pressure—relying on a single station to represent an entire grid cell can introduce significant uncertainty in complex terrain.







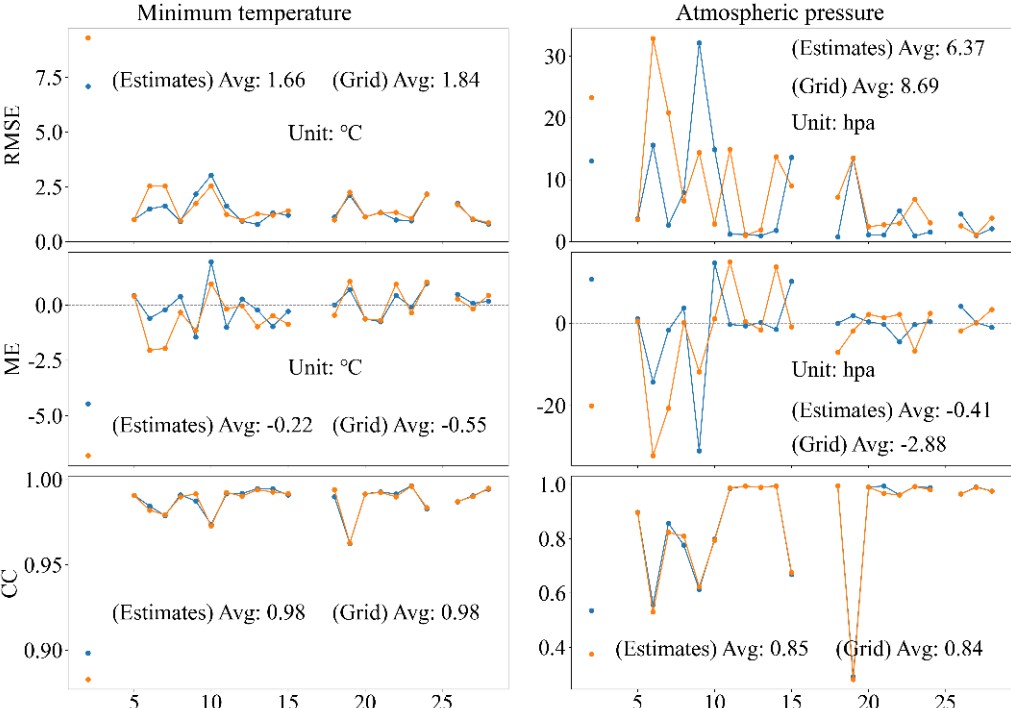

**Figure 10: Comparison of dotted line plots for RMSE, ME, and CC between in-situ data and station-based estimates, as well as between in-situ data and gridded data.**

**4.5 Spatial distribution of meteorological elements in China at 1 km resolution**

To evaluate the spatial performance and climatic representativeness of the reconstructed dataset, we analyzed the long-term mean values of six meteorological variables at a spatial resolution of 1 km across mainland China from 1961 to 2019. The spatial distributions show strong consistency with known climatic gradients and topographic variations, reflecting the combined effects of latitude, elevation, and oceanic influence on regional meteorological conditions, as illustrated in Figure 11. Temperature exhibits clear spatial variation governed by both latitude and elevation. The Northern Temperate Zone and the Plateau Climate Zone record the lowest values, with annual mean, maximum, and minimum temperatures of −3.8 °C, 4.3 °C, and −11.0 °C in the Northern Temperate Zone, and −1.7 °C, 6.2 °C, and −8.3 °C in the Plateau Climate Zone. In contrast, the Subtropical Zone records 16.1 °C, 21.3 °C, and 12.5 °C, while the Tropical Zone reaches 24.2 °C, 28.9 °C, and 21.1 °C, respectively. Atmospheric pressure strongly reflects elevation differences. While most zones maintain annual mean values above 900 hPa, the Plateau Climate Zone shows a significantly lower pressure of approximately 608 hPa. Relative humidity decreases from southeast to northwest, shaped by maritime influence and topographic relief. The Tropical and Subtropical coastal zones record the highest annual mean values of 83 % and 78 %, respectively. The Northern Temperate Zone reaches 70 %, while interior zones, including the Middle Temperate and Plateau Climate Zones, record lower values of approximately 55 %. Sunshine duration shows an inverse pattern relative to humidity and cloudiness. The longest annual

average sunshine durations are observed in the Qinghai–Tibet Plateau and the Middle Temperate Zone in Xinjiang and Inner Mongolia, with 8.0 h and 7.8 h per day, respectively. In contrast, the Subtropical coastal zone receives only about 4.6 h due to persistent cloud cover and high moisture levels.

The reconstructed spatial patterns show strong agreement with China's climatic zonation and physiographic structure, demonstrating that the dataset reliably captures the spatial distribution of key climate-controlling factors, including elevation, latitude, and terrain complexity. This consistency highlights the physical soundness and regional adaptability of the reconstruction framework, which is informed by topographic features rather than relying solely on spatial proximity. The dataset thereby offers robust support for regional-scale analyses in hydrology, meteorology, and ecology, especially in

contexts where high spatial resolution and internal data consistency are required.

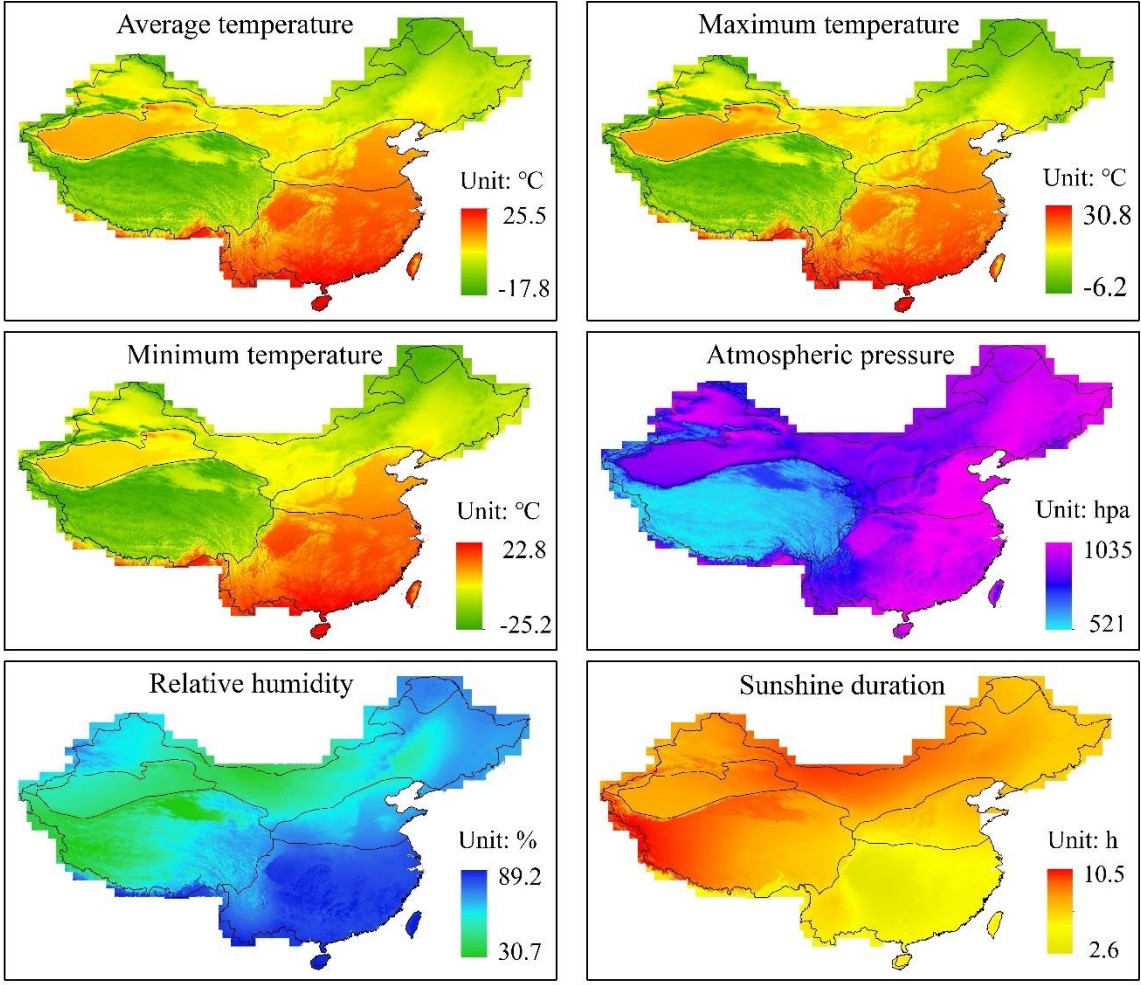

**Figure 11: Annual spatial distribution of 6 meteorological elements in China from 1961 to 2019 based on daily reconstructed products.**



## 5 Data availability

The 1 km daily dataset of near-surface meteorological variables over mainland China includes air temperature (average, maximum, and minimum) for the period 1961–2021, and atmospheric pressure, relative humidity, and sunshine duration for the period 1961–2019. The dataset is expected to undergo ongoing maintenance and temporal extension contingent on the availability of new observational data. The GeoTIFF-formatted output files at 1 km spatial resolution are freely accessible at https://doi.org/10.11888/Atmos.tpdc.301341 (Zhao et al., 2024).

## 450 6 Conclusion

This study presents a nationwide, high-resolution dataset of six daily near-surface meteorological variables—average, maximum, and minimum temperature, atmospheric pressure, relative humidity, and sunshine duration—reconstructed at 1 km spatial resolution over mainland China for the period 1961–2019 (1961–2021 for air temperature). Instead of relying on traditional spatial interpolation, the reconstruction framework models nonlinear relationships between meteorological

variables and topographic predictors—such as elevation, latitude, and longitude—enabling physically informed estimation across a wide range of climatic and geographic conditions.

Validation using 118 independent meteorological stations demonstrates that the dataset achieves consistently high accuracy across all variables. For average, maximum, and minimum temperature, the median RMSEs are 1.03 °C, 1.19 °C, and 1.34 °C, respectively; the corresponding median MEs are approximately −0.09 °C, −0.10 °C, and −0.08 °C, with

correlation coefficients equal to or greater than 0.99. Atmospheric pressure shows similarly strong performance, with a median RMSE of 2.48 hPa, a median ME of −0.02 hPa, and a correlation coefficient of 0.98. Relative humidity and sunshine duration also perform reliably, with median RMSEs of 6.02% and 1.48 h, MEs of −0.5% and 0.05 h, and correlation coefficients of 0.90 and 0.93, respectively. Further comparison reveals that station-to-grid validation underestimates the true accuracy of gridded products, particularly in topographically complex regions where elevation mismatches distort point-to-

grid comparisons. In such areas, model estimates based on exact station coordinates consistently yield better validation metrics than those derived from station-to-grid comparisons, especially for elevation-sensitive variables.

The comparative evaluation against existing gridded products further confirms the quality and robustness of the reconstructed dataset, while complementing existing benchmark products with enhanced spatial resolution (1 km), particularly suited for heterogeneous environments. For average temperature, atmospheric pressure, and relative humidity,

the reconstructed product exhibits consistently lower RMSE and ME than CMFD 2.0 at independent validation stations, with particularly substantial error reduction observed for atmospheric pressure. In the comparison of sunshine duration, the reconstructed dataset achieves nearly identical accuracy to the SSD benchmark product in terms of RMSE, ME, and CC, despite being validated using stations that may have been assimilated during the development of the SSD product.

In addition to its high overall accuracy, the dataset demonstrates stable spatial performance across China's major

climatic zones. Temperature and pressure variables maintain low RMSEs and strong correlations in both humid southeastern



and arid northwestern regions, with most temperature RMSEs, MEs, and CCs falling within the ranges of 0.49 °C to 2 °C, −2 °C to 2 °C, and 0.95 to 1.00, respectively. Relative humidity and sunshine duration show limited spatial variability, with only a few isolated stations displaying notable deviations. Even in data-sparse regions like Taiwan—excluded from model training—the reconstructed temperature fields align reasonably well with in-situ observations, indicating the dataset's spatial robustness beyond the training domain.

The dataset provides spatially continuous, temporally complete, and variable-accurate daily meteorological records, supporting a wide range of regional-scale applications in hydrology, meteorology, and ecology.

**Author contributions.** KZ: methodology, conceptualization, formal analysis, visualization, and writing – original draft. DY and TQ: supervision, methodology, and writing – review and editing. CL and DP: software, formal analysis, and investigation. YS: data curation, visualization.

**Competing interests.** The contact author has declared that none of the authors has any competing interests.

**Acknowledgements.** We would like to express our sincere gratitude to Alibaba Cloud for providing high-performance computing support. We also extend our thanks to the National Meteorological Information Center of the China Meteorological Administration for supplying the observed climate data.

**Financial support.** This work was supported by the National Natural Science Foundation of China (Grant No. 52130907), the Postdoctoral Fellowship Program of the China Postdoctoral Science Foundation (Grant No. GZC20233116), and the Five Major Excellent Talent Programs of IWHR (Grant No. WR0199A012021).

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
