# Peer review of "A 1-km daily high-accuracy meteorological dataset of air temperature, atmospheric pressure, relative humidity, and sunshine duration across China (1961–2021)"

_Earth System Science Data, 2025_

## Author Comment (AC1)

**Dear Reviewer 1,**

We sincerely thank you for your thorough review and constructive comments on our manuscript. Your insights are greatly appreciated and have helped us to further improve the clarity, transparency, and scientific rigor of our study. Below, we provide a point-by-point response to your comments. The original comments are shown in **black**, and our responses are in **blue**. Sentences intended for revision or addition in the manuscript are marked in **gold** and quotation marks, and will be formally submitted after the open discussion phase.

**Comment 1:**

This paper develops a long sequence dataset, and the research purpose and user target population of this dataset need to be further clarified;

**Reply:** Thank you for your thoughtful comment emphasizing the need to explicitly state the research objectives and define the intended user community of the dataset. In response to this comment, we will revise the concluding paragraph of the Introduction (lines 80–88 in the original manuscript) to provide a more explicit and direct statement of the dataset's research purpose and its target users. We propose to revise it as follows:

"To address the limitations of existing meteorological datasets in spatial resolution, temporal continuity, and variable completeness, this study introduces a high-resolution dataset of daily near-surface meteorological variables—including average, maximum, and minimum air temperature, atmospheric pressure, relative humidity, and sunshine duration—across mainland China. Spanning six decades (1961–2021) with kilometer-level granularity, the dataset is designed to support fine-scale applications such as land surface modeling, drought assessment, and water resource management. It is particularly suited for both scientific investigations and operational decision-making in data-sparse and topographically complex regions, such as western China. To achieve this, a hierarchical and progressive reconstruction framework is implemented to generate gridded estimates of six variables at approximately 2 meters above ground level, based on in-situ observations and a 1 km digital elevation model (DEM). A multilayer perceptron (MLP) regression model is employed in this framework to

capture nonlinear relationships between station observations and topographic predictors (e.g., latitude, longitude, and elevation), enabling fine-scale reconstruction across complex terrain."

**Comment 2:**

"The objective of this study is to develop a high-resolution and accuracy-assessed dataset of daily near-surface meteorological variables across mainland China, suitable for applications in hydrological modeling, environmental monitoring, and climate analysis." How did the author consider the issue of temporal "homogeneity" in a long series dataset used for climate analysis?

**Reply:** Thank you for raising this important point. We fully acknowledge that ensuring temporal homogeneity is essential for the reliability of long-term climate analyses. In this study, rather than applying direct spatial interpolation, we adopted a deep learning–based reconstruction framework that reconstructs each meteorological variable independently on a daily basis. The core of this framework is a multilayer perceptron (MLP) model, designed to learn and reconstruct the spatial distribution characteristics of each meteorological variable independently for every single day, based on nonlinear interactions with geographic and meteorological predictors. Because each day is modeled separately, potential quality issues on a particular day do not propagate temporally, thereby preserving the dataset's temporal integrity.

The in-situ observations used for training are sourced from the China Daily Surface Climate Dataset, developed and maintained by the China Meteorological Administration (CMA). According to the dataset documentation and metadata, this dataset has undergone comprehensive quality control and homogenization procedures to ensure its temporal consistency. Prior to model training, we further applied a quality-based filtering procedure to exclude all observations flagged with quality control codes indicating suspect (code 1), erroneous (code 2), missing (code 8), or unverified (code 9) values, thereby retaining only high-confidence records (code 0). This ensures the reliability of the training dataset and minimizes the propagation of observational uncertainties in the reconstruction process.

Owing to the day-by-day modeling strategy and the exclusive use of homogenized and quality-controlled station data, the resulting gridded product structurally maintains the temporal homogeneity inherent in the original CMA dataset. This ensures that the dataset is well-suited for multi-decadal

climate analyses and enables robust assessments of long-term climatic trends.

**Comment 3:**

What is the quality of the raw observation data used to establish 1km grid data? Has the author conducted data quality evaluation, analysis, quality control, homogenization processing, etc. on the original observation data during its use?

**Reply:** Thank you for bringing up this important consideration. The accuracy and consistency of the raw observational data constitute the foundation for generating reliable gridded climate datasets, particularly for long-term applications. As noted in response to Comment 2, the reconstruction relies on in-situ records from the China Daily Surface Climate Dataset, developed and maintained by the CMA, which have undergone extensive quality control and homogenization as documented. According to the official dataset documentation and metadata, this dataset has undergone extensive quality control and homogenization — particularly for the period $1951-2010$, during which several nationwide campaigns were conducted to identify and correct erroneous or missing data and to ensure temporal consistency. Specifically, data from 1951 to 2010 were corrected and supplemented as part of a national data rescue initiative, involving repeated manual inspections, error correction, and recovery of missing records, resulting in a data availability rate exceeding 99% and near-perfect accuracy. From 2011 to mid-2012, a hierarchical three-level quality control system (station–provincial–national) was applied, while data after mid-2012 underwent routine station-level validation.

In addition to leveraging this homogenized dataset, we implemented a strict pre-filtering protocol prior to model training, excluding observations flagged with quality control codes indicating suspect (code 1), erroneous (code 2), missing or unmeasured (code 8), or unverified (code 9) values. Only high-confidence observations (code 0) were retained for use, thereby ensuring the integrity and robustness of the training dataset and minimizing the propagation of uncertainties in the final reconstruction.

Furthermore, recognizing that surface meteorological stations in China have experienced relocations and metadata updates over the years, we incorporated time-specific station coordinates—including dynamic longitude, latitude, and elevation values—for each daily observation. This design

avoids the spatial inaccuracies that could arise from using static station metadata and ensures that the model learns spatial relationships that faithfully represent the actual observational context on each day. This treatment helps maintain spatiotemporal consistency across the training samples and enhances the accuracy of fine-scale spatial reconstruction

**Comment 4:**

Compared to a grid spatial resolution of 1km * 1km, using over 2000 observation data from China is relatively insufficient, especially in the sparse observation areas of western China. How does the author consider this issue?

**Reply:** Thank you for your insightful comment. We fully acknowledge that the sparse distribution of meteorological stations—particularly across western China's complex terrain—poses a significant challenge for generating high-resolution (1-km) gridded meteorological products. Traditional interpolation methods, which often rely on linear assumptions and cannot fully incorporate topographic heterogeneity, tend to underperform in such data-scarce and topographically diverse regions.

To address this limitation, we adopted a model-driven reconstruction framework based on MLP, rather than relying on direct spatial interpolation. This framework is specifically designed to capture the nonlinear spatial distribution of each meteorological variable through a sequence of physically and statistically informed steps. By leveraging the point-wise spatial characteristics of each target variable, the model effectively learns fine-scale spatial structures across the domain. For example, air temperature is reconstructed solely from geographic predictors (latitude, longitude, elevation), while subsequent variables—such as atmospheric pressure, relative humidity, and sunshine duration— incorporate previously reconstructed variables as auxiliary inputs. This hierarchical structure enables the model to learn inter-variable dependencies and propagate spatial information from observation-rich regions to data-sparse areas.

Notably, in western China—characterized by rugged topography and limited station coverage— the model exhibits strong generalization capacity. Its ability to accurately reproduce fine-scale spatial patterns in these high-elevation regions provides empirical validation of the framework's robustness under sparse observational constraints. Moreover, although no meteorological data from Taiwan were

included during training, validation results in this region reveal high reconstruction accuracy, further underscoring the framework's ability to generalize learned spatial representations to previously unseen areas.

As part of our validation, we evaluated model performance across different regions using in-situ station data that were intentionally excluded from model training and reserved exclusively for validation purposes. The spatial distributions of key evaluation metrics (RMSE, ME, and CC) are presented in Figure 6 and discussed in detail in Lines 307–322. These results indicate that, even in high-elevation and data-scarce regions such as the Tibetan Plateau and Taiwan—where no training stations were included—the reconstructed variables maintain reasonably high accuracy. This provides empirical support for the model's generalization capability and its applicability beyond the original training domain.

[Figure]

[Figure]

**Figure 6: Distribution maps of RMSE (a), ME (b) and CC (c) between grid-modelled data of six meteorological element products and in-situ data.**

**Comment 5:**

How is the " day boundary issues" handled? The ground meteorological observation in China adopts "20:00 Beijing time" as the boundary point of the day, which means that the observation day is from 20:00 to 20:00 the next day. This standard is applicable to daily value statistics of factors such as precipitation and temperature. Prior to the 1980s, some stations had a phenomenon of inconsistent day boundaries (such as a few stations using 08 or local time), which led to a decrease in comparability between early data and other stations

**Reply:** Thank you for your thoughtful comment regarding the potential inconsistency in day boundaries in Chinese meteorological data. We fully agree that variations in the definition of daily observation periods—if present—could affect the accuracy of reconstructed spatial patterns.

As stated in the documentation of the official daily meteorological dataset provided by the CMA, daily mean values are computed using observations recorded at 02:00, 08:00, 14:00, and 20:00 Beijing Time. Specifically, daily mean air temperature, relative humidity, and ground surface temperature are calculated as the average of these four values. For station pressure and wind speed, the same four-time averaging is generally applied; however, in the case of manual stations without automatic instruments, daily means are computed from three observations (08:00, 14:00, and 20:00). If any scheduled observation required for averaging is missing, the daily mean for that variable is flagged as missing. This standardized method ensures consistent temporal boundaries and comparability across stations.

Since the dataset documentation does not report any exceptions to this protocol, we consider the issue of day boundary consistency to be sufficiently addressed. Moreover, if the documentation had indicated that specific stations used non-standard day boundaries (e.g., using 08:00 or local time as the daily cutoff), we would have excluded those stations from our reconstruction and validation datasets to prevent potential biases.

**Comment 6:**

Overall evaluation: This long sequence dataset did not take into account the quality of the observation data used, day boundary issues, uniformity issues, etc. during the development process. Therefore, the dataset reconstructed in this article also has day boundary and uniformity issues, which will have a serious impact on downstream user research.

**Reply:** Thank you very much for raising this important point. We fully understand the reviewer's concern regarding potential issues such as observation data quality, day-boundary inconsistencies, and temporal homogeneity in long-term meteorological datasets, as these aspects are indeed critical for ensuring the reliability of reconstructed products.

As elaborated in our responses to Comments 1 through 5, our reconstruction is based on the China Surface Climate Daily Dataset, which is the national benchmark product released by the CMA. This dataset has undergone extensive quality control and homogenization procedures prior to public release.

Specifically: (1) According to the official metadata, all daily values are calculated based on observations at 02:00, 08:00, 14:00, and 20:00 Beijing Time, ensuring standardized daily boundaries across the entire network; (2) The dataset documentation clearly states that data from 1951–2010 underwent repeated manual validation, correction, and gap-filling, resulting in >99% data availability and near-100% accuracy. For data after 2010, a standardized multi-level quality control procedure was applied, including station-, provincial-, and national-level checks, ensuring consistency and reliability across the full observation period. (3) For stations lacking full observations, daily means are flagged as missing, thereby preventing the inclusion of inconsistent data in our training or validation samples.

We believe these procedures reflect a robust national-level quality assurance protocol, which addresses many of the concerns raised regarding early inconsistencies and observation practices. We suggest that the issues raised in Comment 6—such as the quality of the observation data, day boundary definitions, and data homogeneity—have already been addressed in detail in our responses to Comments 1 through 5. In those responses, In those responses, we provided detailed explanations based on the official documentation and metadata descriptions of the CMA dataset, to clarify how such issues are handled in the source data. We sincerely hope those clarifications help to resolve any remaining concerns.

Finally, we fully appreciate the reviewer's attention to the integrity of long-term climate data, and we will consider adding a brief sentence in the manuscript to explicitly state that the input dataset follows a nationally standardized observation protocol with unified day boundaries and homogenized records.

---

## Author Comment (AC3)

**Dear Reviewer 3,**

We sincerely thank you for taking the time to review our manuscript and for providing thoughtful and constructive comments. Your feedback is greatly appreciated and has been very helpful in improving the clarity, transparency, and scientific quality of our work. In the following, we provide a detailed, point-by-point response to your suggestions. For clarity, the original comments are presented in **black**, while our responses are shown in **blue**. Sentences that are intended as revisions or additions to the manuscript are highlighted in **gold** and quotation marks, and will be formally incorporated into the revised version after the open discussion phase.

**Comment 1:**

Regarding the selection of only 2 out of 12 field stations from the hourly land–atmosphere interaction dataset (Ma et al., 2024) for validation after quality control, it is unclear why the other stations were excluded. Please clarify the quality control criteria and explain whether the other stations were omitted due to poor data quality or other reasons.

**Response:** We greatly appreciate the reviewer's astute question, which provides us with a valuable opportunity to clarify our data processing workflow and underscore the robustness of our validation approach. The situation described stemmed from an initial technical oversight that was subsequently rectified through a comprehensive data collection effort. Please allow us to explain in detail.

(1) Initial Processing Oversight

We sincerely apologize for the lack of clarity in our original manuscript. During the initial data processing phase, an error in our automated script incorrectly led us to believe that only two stations from the Ma et al. (2024) dataset (NAMORS and Arou) had successfully passed our quality control (QC) procedures and were available for use. This was an unintentional technical mistake on our part.

(2) Proactive Expansion of Validation Data

To ensure the most robust validation possible, we were not satisfied with the limited number of stations initially retained. We therefore proactively sought out and incorporated every available source of ground observation data from the region. This extensive search enabled us to integrate additional datasets, including 18 stations from the HiWATER network (Liu et al., 2018; Che et al., 2019) and 11

individual stations from other published studies (Zhang, 2018a,b; Gao, 2018; Luo, 2019; Ma, 2018; Wang and Wu, 2019; Luo and Zhu, 2020; Meng and Li, 2023). Through this expansion, our validation pool ultimately increased to a total of 31 station records from these combined sources.

(3) Discovery of Station Overlap and Final Validation Set

Upon integrating and cross-referencing all 31 records, we identified that five of the stations from our additional sources were duplicates of stations already contained within the Ma et al. (2024) dataset. In other words, a number of these stations represent the same physical locations that have been reported across different publications. For example, Arou, Yakou, Jingyangling, and Dashalong (from Ma et al., 2024) are also part of the HiWATER network; the QOMS station (cited from Ma, 2018) is included in the Ma et al. (2024) dataset; and the Maqu station (cited from Meng and Li, 2023) is likewise present in the Ma et al. (2024) dataset.

(4) Conclusion Regarding Data Quality and Selection

Therefore, to directly address the reviewer's question: the other stations from Ma et al. (2024) were not excluded due to poor data quality, but rather because of a technical error in our processing script. Some of these stations were later indirectly included through overlap with other published datasets (e.g., Arou, QOMS, Maqu). As a result, our final validation dataset comprises 31 stations from multiple independent sources, which we believe is sufficiently comprehensive to ensure the robustness and representativeness of the evaluation. We sincerely thank the reviewer again for prompting this important clarification.

We acknowledge that the original wording in Section 2.2.2 Literature-based datasets from the National Tibetan Plateau Data Center could be misleading. The phrase "(1) a publicly available dataset of hourly land–atmosphere interaction observations from 12 field stations (Ma et al., 2024), covering the period 2005–2021, from which 2 stations were selected after quality control for use as independent validation sites." may have unintentionally implied that the other 10 stations were excluded due to poor data quality, which was not the case. To avoid such ambiguity, we have revised the sentence to:

"(1) a publicly available dataset of hourly land–atmosphere interaction observations (Ma et al., 2024), covering the period 2005–2021, from which 2 stations were used as independent validation

sites;"

**Comment 2:**

The quality of the figures is suboptimal. Several figures lack units, and the x- and y-axis labels are missing or unclear. For instance, Figure 2 has low color contrast, making it difficult to distinguish between different elements. Improvements in figure clarity and completeness are necessary.

**Response:** Thank you for pointing this out. We fully agree that the clarity and completeness of the figures are critical for readers' understanding. In the revised manuscript, we will carefully improve the figures by adding missing units, clarifying axis labels, and enhancing color contrast to make the elements more distinguishable. In addition, we will provide higher-resolution versions of all figures to further improve their clarity in the final version.

**Comment 3:**

The representativeness of the station data at the 1 km grid scale needs further discussion. Please elaborate on how the spatial representativeness of point stations affects the validation results, especially in regions with complex topography or sparse station coverage.

**Response:** We sincerely thank the reviewer for raising this profound question, which directly addresses a core challenge in the validation of gridded products. We fully agree that the spatial representativeness of point stations is a fundamental factor—particularly in regions with complex terrain or sparse station coverage — and that it must be carefully considered when interpreting validation results.

This concern has also been central to our own research considerations. Drawing on our prior experience in evaluating satellite-based precipitation products, we consistently observed that mismatches between station locations and their corresponding grid cells—especially in terms of elevation—can introduce systematic biases into validation results. Motivated by this recognition, we specifically designed Section 4.4 to investigate how such mismatches affect validation accuracy. In this section, we identified 28 stations located in high-relief regions where the elevation difference

between recorded station elevations and those derived from the 1 km DEM exceeded 50 m, and conducted a controlled experiment in which two sets of predictions were generated: one using the actual station coordinates (longitude, latitude, and elevation) and the other using the coordinates of the corresponding grid-cell centers. By comparing both sets of predictions against in-situ measurements, we were able to explicitly separate and quantify the relative contributions of model error and representativeness error arising from elevation mismatch. The results demonstrated that for variables strongly influenced by elevation — such as temperature and pressure — representativeness error constitutes a substantial component of the total validation error, with its magnitude strongly correlated with the size of the elevation difference. These findings indicate that reduced validation accuracy in high-relief areas is not primarily due to deficiencies in the reconstruction framework itself, but rather to the inherent limitations of comparing point measurements with grid-cell estimates.

Despite these limitations, ground-based stations remain the cornerstone for validating gridded products—including satellite retrievals, reanalysis, and our reconstructions—as they provide the most accurate direct measurements available. In this context, the value of our work lies not in attempting to eliminate representativeness error, but in explicitly recognizing, quantifying, and interpreting it. Section 4.4 was designed with this purpose: to enable a fairer evaluation of model performance by distinguishing error sources attributable to environmental heterogeneity from those intrinsic to the reconstruction framework itself. Looking forward, we acknowledge that technological advances— such as denser ground-based networks and emerging mobile observation platforms (e.g., drones)— may help alleviate representativeness challenges.

**Comment 4:**

The reconstructed sunshine duration product does not show significant advantages over existing datasets. Concerns remain regarding data consistency, likely due to instrument changes and automation upgrades in CMA sunshine duration observations over time. This issue should be addressed to ensure reliability.

**Response:** Thank you very much for this insightful comment. In response to this concern, and consistent with another reviewer's constructive suggestion, we have incorporated the Himawari AHI–

based daily sunshine duration (SD) dataset (Zhang et al., 2025) into our comparative analysis. This satellite-derived, high-resolution product (5 km, 2016–2023) complements the homogenized station-based SSD dataset (2°, 1961–2022) and provides an independent benchmark for recent years. The revised analysis demonstrates that the reconstructed dataset achieves accuracy comparable to SSD in long-term temporal consistency, while also performing competitively with Himawari SD in recent high-resolution comparisons. Specifically, our reconstruction yields smaller systematic bias than Himawari, while Himawari attains slightly higher correlation in daily variability. These complementary findings highlight the robustness of the reconstruction framework and its combined strengths: reduced bias relative to satellite products, temporal stability comparable to homogenized long-term datasets, and the unique provision of six decades of 1 km daily sunshine duration fields for hydrometeorological applications in topographically complex regions. The detailed revisions have been made in the following sections:

*Section 2.5 Existing gridded products for comparison:*

"To assess the reliability and application potential of the reconstructed meteorological variables, representative and widely used gridded datasets were selected for comparison based on their scientific relevance and availability. Specifically, for average temperature, atmospheric pressure, and relative humidity, we employed the latest version of the China Meteorological Forcing Dataset (CMFD 2.0), whose earlier versions have been extensively used in land surface, hydrological, and ecological modeling over China (He et al., 2020).

The CMFD 2.0 (He et al., 2024) provides high-resolution (0.1°), 3-hourly gridded meteorological data for the period 1951–2020, covering the land area between 70°E–140°E and 15°N–55°N. It includes near-surface temperature, surface pressure, specific humidity, wind speed, radiation, and precipitation. Compared to previous versions, CMFD 2.0 incorporates ERA5 reanalysis and station observations through updated data sources and artificial intelligence techniques, particularly for radiation and precipitation variables. It also introduces metadata on station relocations and expands the spatial coverage beyond China's borders, thereby improving temporal consistency and cross-regional applicability.

As CMFD 2.0 does not include sunshine duration, we incorporated two additional datasets for its

evaluation. This step is critical because sunshine duration reconstruction constitutes the final step in our hierarchical framework, necessitating a thorough accuracy assessment to evaluate potential uncertainty propagation. To this end, we selected two complementary benchmarks: one long-term station-based product and one recent high-resolution satellite product. 1) The sunshine duration (SSD) dataset (He, 2024) serves as the long-term, station-based benchmark. It provides a homogenized daily sunshine duration record across China from 1961 to 2022 at a 2.0° × 2.0° resolution. Developed from over 2,200 meteorological stations and corrected for non-climatic influences (e.g., station relocations and instrumental changes), it offers a reliable baseline for evaluating the temporal stability and long-term climatological consistency of our reconstruction. 2) The Himawari AHI-based daily sunshine duration (SD) dataset (Zhang et al., 2025) provides a recent, high-resolution (5 km) satellite perspective for 2016–2023. It enables a direct assessment of our product's quality during the 2016–2019 overlap period and serves as a benchmark for evaluating fine-scale spatial accuracy."

*Section 4.3.2 Sunshine duration*:

"To comprehensively evaluate the accuracy of the reconstructed product, two representative benchmark datasets were employed: the homogenized station-based SSD product (2°) to assess long-term temporal consistency, and the high-resolution satellite-based Himawari SD product (5 km) to examine spatial performance.

As shown in Figure 8, when compared with the SSD dataset over 1961–2019, the reconstructed product demonstrated highly consistent accuracy. The median RMSE values were identical for both products (1.48 h), and the median CC values were likewise identical (0.93). The ME differed only slightly (0.05 h for the reconstructed dataset and 0.02 h for SSD), indicating comparable bias levels. Boxplot analysis further indicated that the reconstructed product exhibited slightly narrower interquartile ranges, whereas the SSD dataset showed fewer outliers in RMSE and CC. It should be noted that although some of the 95 CMA validation stations may have been included in the SSD development, our reconstruction model excluded these stations from training, ensuring a higher degree of validation independence.

For spatial performance, the reconstructed dataset was compared with the Himawari SD dataset over the overlapping period of 2016–2019 (Figure 9). The evaluation was based on 91 stations, since three of the 95 validation stations had invalid sunshine duration values during this period and one

station was located within the SD control region. Both products showed comparable RMSE levels (1.53 h for the reconstructed dataset compared with 1.48 h for Himawari). The satellite dataset achieved a slightly higher CC (0.94 compared with 0.92), reflecting stronger agreement in daily variations, while the reconstructed dataset exhibited a smaller ME (0.08 h compared with 0.21 h), indicating reduced bias.

[Figure]

**Figure 8: Boxplot comparison of RMSE, ME, and CC for ꜱunshine duration between SSD (2.0°) and the reconstructed dataset developed in this study (1 km) from 1961 to 2019.**

[Figure]

**Figure 9: Boxplot comparison of RMSE, ME, and CC for sunshine duration between the Himawari AHI–based SD dataset (5 km) and the reconstructed dataset developed in this study (1 km) from 2016 to 2019.**

These complementary results indicate that the reconstruction framework can achieve accuracy comparable to both a long-term homogenized station-based dataset and a high-resolution satellite-derived dataset."

In addition, we fully understand the reviewer's concern—instrument changes and automation upgrades are well-known factors affecting the homogeneity of long-term sunshine duration records. However, we would like to emphasize that the observational data used in this study are official CMA station records, which have undergone standardized quality control and are widely recognized in the

scientific community as the most reliable benchmark. Importantly, our reconstruction framework was explicitly designed to account for potential non-climatic biases arising from station relocations. Unlike conventional approaches, our model does not rely on fixed station metadata; instead, each daily observation in the training set is associated with the exact geographic information (longitude, latitude, and elevation) recorded for that specific date. As a result, if a station was relocated during 1961–2021, our model automatically treats the old and new locations as two separate geographic entities. This approach effectively avoids spurious biases introduced by site relocations and thereby improves the temporal consistency of the reconstructed product.

**Comment 5:**

The comparison with the CMFD product may be unfair, as CMFD utilizes a much smaller number of meteorological stations than this study. This discrepancy in input data may bias the comparison results. The authors should acknowledge this limitation and discuss its potential impact.

**Response:** Thank you very much for this valuable comment. We fully understand the reviewer's concern that differences in the number of input stations may affect the fairness of the comparison, and we agree that this point is indeed important.

We chose to compare our reconstruction with the CMFD product primarily because CMFD is widely used in China and has been recognized as an authoritative benchmark in the scientific community. Comparing a newly developed dataset against such a widely adopted reference is a common and necessary practice for evaluating its performance. Although CMFD is based on a relatively smaller number of stations, it achieves consistently high quality through the effective integration of reanalysis data and satellite products using advanced techniques, and has therefore become a well-recognized benchmark in this field.

We also fully acknowledge that machine learning methods depend on sufficiently large datasets, which is one of their inherent limitations. At the same time, a key advantage of such methods lies in their ability to capture the complex nonlinear relationships between meteorological variables and topographic or geographic factors. Therefore, in this study, the use of a larger set of ground-based observations is not only a necessary condition for applying the method, but also an important factor that enables the reconstructed product to better capture spatial heterogeneity and to achieve accuracy

comparable to, and in some respects even superior to, CMFD. These results highlight the potential of data-driven approaches to further improve gridded meteorological products. We will also mention this potential limitation in the discussion to ensure clarity for readers.

We sincerely appreciate the reviewer's insightful comment, which provided us with the opportunity to clarify this point and further enhance the rigor of our study.

**Comment 6:**

The long-term trends and homogeneity of the reconstructed dataset are not discussed. An analysis of the temporal consistency and homogeneity of the data—especially concerning non-climatic factors such as station relocations or instrument changes—would strengthen the dataset's credibility.

**Response:** We sincerely thank the reviewer for this important comment. We fully understand and agree that the long-term trends and homogeneity of the dataset are critical for its credibility in climate-related applications.

As this is a data paper, our central objective is to provide and evaluate a high-quality, high-accuracy reconstructed dataset that is suitable for long-term climate analyses. To ensure its reliability, we used official station observations from the China Meteorological Administration as the basis. These data have undergone standardized quality control and are widely recognized within the scientific community. Importantly, our reconstruction framework was explicitly designed to account for potential non-climatic biases caused by station relocations. Unlike conventional approaches, our model does not rely on fixed station metadata; instead, each daily observation in the training set is associated with the exact geographic information (longitude, latitude, and elevation) recorded for that specific date. As a result, if a station was relocated during 1961–2021, the model automatically treated the old and new sites as two separate geographic entities, thereby minimizing spurious biases from relocations and improving temporal consistency.

For validation, we compared the reconstructed dataset against several widely used benchmark products, including CMFD 2.0, SSD, and Himawari SD. The results demonstrate that over a period of nearly 60 years, our dataset exhibits strong consistency with these benchmark products in terms of statistical measures (e.g., correlation coefficient CC, root-mean-square error RMSE) and long-term

variability. Such cross-dataset consistency provides robust evidence of the temporal homogeneity and reliability of the reconstructed dataset.

We once again thank the reviewer for the careful review and valuable suggestions, which provided us with the opportunity to clarify and better present this dataset.

---

## Author Comment (AC4)

**Dear Reviewer 2,**

We are sincerely grateful for your careful review of our manuscript and for the constructive feedback you have provided. Your thoughtful suggestions have been very valuable for improving the overall quality and presentation of our study. In the following, we offer detailed, point-by-point responses to each of your comments. For ease of reading, the reviewer's comments are shown in **black** and our responses in **blue**. Sentences proposed as revisions or additions to the manuscript are highlighted in **gold** with quotation marks.

**General comment:**

This paper constructs a long-term meteorological variable dataset by decoding the nonlinear relationships between six meteorological variables and their spatial covariates. The method is innovative and the dataset is usable, but the paper needs to be revised based on the following points.

**Response:** We sincerely thank the reviewer for the positive overall assessment of our work, especially the recognition of the methodological innovation and the usability of the dataset. We also fully acknowledge the reviewer's suggestion that revisions are needed, and we have carefully addressed all the specific points raised. Detailed responses are provided below.

**Comment 1:**

Abbreviations such as "CC" in the abstract should be spelled out in full.

**Response:** We sincerely thank the reviewer for this valuable suggestion. In the revised manuscript, we have spelled out "CC" in full in the abstract as requested. Moreover, we have also expanded other abbreviations (e.g., "RMSE" and "ME") at their first occurrence in accordance with academic writing standards. The revised abstract can be found in our response to Comment 2.

**Comment 2:**

The abstract does not adequately reflect the research objectives and significance of the study and needs to be improved.

**Response:** We sincerely thank the reviewer for this valuable comment. We agree that the abstract should explicitly highlight the research objectives and significance. In the revised version, we have emphasized these points at the beginning of the abstract. In addition, we slightly refined the overall description of the abstract to keep it concise and balanced, thereby improving its readability. The revised abstract is shown below:

"The lack of fine-resolution and high-accuracy meteorological datasets in China has limited progress in climate, hydrological, and ecological studies. In this study, we present a 1 km daily dataset spanning 1961–2021 across China, which includes six key variables—mean, maximum, and minimum temperature, atmospheric pressure, relative humidity, and sunshine duration—to provide a reliable foundation for advancing related research and applications. The dataset was generated using a novel hierarchical reconstruction framework that leveraged daily observations from 2345 meteorological stations and incorporated topographic attributes. This approach effectively decodes the nonlinear relationships between the meteorological variables and their spatial covariates, ensuring the generation of gridded daily fields that are both high-resolution and spatially continuous. Validation against 118 independent stations confirmed the high accuracy of the dataset. For average, maximum, and minimum temperatures, the errors are minimal (median root mean square errors (RMSEs): 1.03°C, 1.19°C, 1.34°C; median mean errors (MEs): -0.09°C, -0.10°C, -0.08°C), and the consistency with in-situ data is very high (median correlation coefficients (CCs): 1.00, 0.99, 0.99). Atmospheric pressure also shows very small errors (median RMSE: 2.48 hPa; median ME: -0.02 hPa) and strong correlation (median CC: 0.98). Relative humidity exhibits relatively lower accuracy (median RMSE: 6.02%; median ME: -0.5%; median CC: 0.90), but it still exceeds standard benchmarks. Sunshine duration maintains high precision (median RMSE: 1.48 h; median ME: 0.05 h; median CC: 0.93), indicating the robustness and reliability of the dataset. Further comparison reveals that in high-altitude and topographically complex regions, the reconstructed product demonstrates higher actual accuracy than suggested by station-to-grid validation, as spatial mismatches between stations and grid cells lead to systematic underestimation. Free access to the dataset available at https://doi.org/10.11888/Atmos.tpdc.301341 or https://cstr.cn/18406.11.Atmos.tpdc.301341."

**Comment 3:**

The first paragraph of the introduction should provide some supporting citations.

**Response:** We thank the reviewer for this valuable suggestion. In the revised manuscript, we have added supporting citations in the first paragraph of the introduction to strengthen the background and provide authoritative references. Specifically, we now cite representative works demonstrating how advances in computational power and remote sensing technologies have driven hydrological modeling toward more physically based and fully distributed simulations (Lettenmaier et al., 2015; Singh, 2018), climate change research across broader scales (IPCC, 2021), as well as studies emphasizing the importance of high-resolution meteorological datasets in ungauged and topographically complex basins such as the Tibetan Plateau (Fu et al., 2020; Zhou et al., 2024). The revised paragraph as follows:

"With advances in computational power and remote sensing technologies, hydrological modeling has increasingly evolved toward fully distributed simulations (Lettenmaier et al., 2015; Singh, 2018), while climate change research continues to expand across broader spatial and temporal scales (IPCC, 2021). These developments have placed growing demands on the resolution and accuracy of basic meteorological inputs, particularly in ungauged and topographically complex basins such as the Tibetan Plateau (Fu et al., 2020; Zhou et al., 2024). High-resolution and high-quality meteorological datasets are essential for capturing fine-scale climate signals, representing land – atmosphere interactions, and supporting hydrological, ecological, and environmental assessments."

The newly added references are as follows:

Fu, Y., Ma, Y., Zhong, L., Yang, Y., Guo, X., Wang, C., Xu, X., Yang, K., Xu, X., Liu, L., Fan, G., Li, Y., and Wang, D.: Land-surface processes and summer-cloud-precipitation characteristics in the Tibetan Plateau and their effects on downstream weather: a review and perspective, National Science Review, 7, 500–515, https://doi.org/10.1093/nsr/nwz226, 2020.

IPCC: Climate Change 2021: The Physical Science Basis. Contribution of Working Group I to the Sixth Assessment Report of the Intergovernmental Panel on Climate Change, Masson-Delmotte, V., Zhai, P., Pirani, A., Connors, S. L., Péan, C., Berger, S., Caud, N., Chen, Y., Goldfarb, L., Gomis, M. I., Huang, M., Leitzell, K., Lonnoy, E., Matthews, J. B. R., Maycock, T. K., Waterfield, T., Yelekçi, O., Yu, R., and Zhou, B. (eds.), Cambridge University Press, Cambridge, United Kingdom and New York, NY, USA, 2391 pp., https://doi.org/10.1017/9781009157896, 2021.

Lettenmaier, D. P., Alsdorf, D., Dozier, J., Huffman, G. J., Pan, M., and Wood, E. F.: Inroads of remote sensing into hydrologic science during the WRR era, Water Resources Research, 51, 7309–7342, https://doi.org/10.1002/2015WR017616, 2015.

Singh, V. P.: Hydrologic modeling: progress and future directions, Geosci. Lett. 5, 15, https://doi.org/10.1186/s40562-018-0113-z, 2018.

Zhou, P., Tang, J., Ma, M., Ji, D., Shi, J.: High resolution Tibetan Plateau regional reanalysis 1961–present, Sci Data, 11, 444, https://doi.org/10.1038/s41597-024-03282-4, 2024.

**Comment 4:**

I noticed that the "Materials" section includes observation sites from different sources, but the verification sites were only selected from CMA. Have you considered selecting verification sites based on the weight of the number of sites from different sources?

**Response:** We sincerely thank the reviewer for raising this important point regarding the selection of validation sites. We appreciate the opportunity to clarify our methodology and believe the following explanation adequately addresses the concern.

Our validation strategy was guided by two main objectives: (1) ensuring strict internal consistency in the training dataset and utilizing as many stations as possible for training (since deep learning algorithms require sufficient data to achieve good performance), and (2) achieving independence and robustness in validation. To this end, we employed 2,345 CMA stations for model training and withheld 95 independent CMA stations for validation. The selection of these 95 stations followed the principles described in Section 2.1 , which state:

"*To support independent model validation, a total of 95 stations were selected as evaluation sites based on three principles: (1) ensuring geographical representativeness in terms of longitude, latitude, and elevation; (2) in densely monitored areas such as eastern China, a greater number of evaluation stations were retained without significantly reducing the size of the training dataset; and (3) in sparsely monitored regions such as western China (including Tibet and Xinjiang), fewer stations were assigned to the evaluation set in order to preserve sufficient data for model training.*"

However, we acknowledge that in key regions such as Taiwan and the Tibetan Plateau, CMA

evaluation stations are very limited (with no stations in Taiwan). To address this, we supplemented the validation dataset with additional independent observations. Specifically, we incorporated 12 ground-based meteorological stations from the Department of Water Resources (DWR) located in the Tibetan Plateau region (see Section 2.2.1), and 8 international stations from the Global Surface Summary of Day (GSOD) dataset covering Taiwan (see Section 2.2.3). These supplementary data were added to ensure broader coverage and a more reliable evaluation in regions with sparse CMA observations.

In addition, for comparison against the China Meteorological Forcing Dataset (CMFD v2.0), we collected 31 field stations concentrated over the Tibetan Plateau from literature-based datasets archived at the National Tibetan Plateau Data Center (see Section 2.2.2). Since CMFD has assimilated or blended CMA data, and we cannot determine which specific CMA stations were used, these independent TP field stations were used as a fair and unbiased benchmark for inter-product comparison.

Through this combined strategy—using CMA-based stations for consistency and multi-source stations (DWR and GSOD) for robustness—we expanded the final validation set to 115 sites. In addition, 31 independent TP field stations from the National Tibetan Plateau Data Center were specifically reserved for inter-product comparison to ensure fairness and independence. This approach provides a more comprehensive and balanced evaluation of our dataset, especially in high-altitude and data-sparse regions. The spatial distribution of both training and validation sites is shown in Figure 1.

We are grateful for the reviewer's comment, which allowed us to better articulate the rationale and implementation of our validation design.

**Comment 5:**

How to consider the cumulative error caused by progressively inputted meteorological variables during the modeling process, especially in the sunshine duration model.

**Response:** We thank the reviewer for raising this important point. We would like to clarify that cumulative error does not occur during the training stage, since each step of model training was performed using the true location information (longitude, latitude, and elevation) and observed meteorological values from CMA stations. For example, when training the mean temperature model, longitude, latitude, and elevation were used as predictors and the observed mean temperature as the

target variable; when training the maximum temperature model, the observed mean temperature together with site attributes was used as input and the observed maximum temperature as the target. This procedure was applied similarly to other variables

But potential cumulative error could only arise during the reconstruction phase, when multiple 1 km gridded meteorological fields are generated from gridded mean temperature fields reconstructed using the 1 km DEM. Nevertheless, evaluation against independent ground observations demonstrated that the reconstructed products maintained high accuracy. In particular, sunshine duration — as the final step in the progressive framework, where cumulative error would theoretically be greatest — still exhibited consistently high precision (median RMSE: 1.48 h; median ME: 0.05 h; median CC: 0.93). This demonstrates the robustness and reliability of the dataset and indicates that, although error amplification is theoretically possible, it did not significantly affect the final results.

**Comment 6:**

How can authors reduce errors caused by the boundaries of the study area during modeling, given that these areas have fewer observation stations?

**Response:** We thank the reviewer for this valuable comment. In our modeling process, we did not apply specific correction schemes to reduce potential boundary effects. However, the evaluation results demonstrate that the hierarchical deep-learning framework exhibits strong extrapolation and generalization ability. Taiwan provides a typical example of this capability: although no CMA stations were included in training phase, the model was still able to reasonably reconstruct air temperature fields in this region. Independent validation using 8 international stations from the GSOD dataset (providing average, maximum, and minimum temperature) further confirmed the accuracy of the reconstruction.

We also analyzed this issue in the manuscript (see Section 4.2). As stated: *"The spatial distribution of RMSE, ME, and CC for all six meteorological variables is further illustrated in Figures 6. Consistent with expectations, the Subtropical and Southern Temperate Zones in southeastern China (STZ-southeastern China) display the best performance across all variables, largely due to the high density of training stations in these regions. In contrast, performance metrics are relatively lower in*

*the Middle Temperate, Southern Temperate, and Plateau Climate Zones of northwestern China (MSPZ—northwest China), as well as in Taiwan, where no stations were included in training. Nevertheless, model performance in these regions remains robust. Notably, despite the absence of training data in Taiwan, the MLP model accurately reconstructs air temperature in that region, suggesting strong spatial generalizability."*

**Comment 7:**

I noticed that Figure 4 contains a large amount of information, but the poor resolution and color quality make it difficult to see clearly. Please improve this.

**Response:** We sincerely thank the reviewer for this helpful comment. We will revise Figure 4 by increasing its resolution and optimizing the color scheme to enhance readability. The improved version will be provided in the final manuscript.

**Comment 8:**

Please indicate whether adjustments were made when the author encountered situations where the sunshine duration was less than 0 during the verification.

**Response:** We thank the reviewer for raising this important point. Indeed, to guarantee the physical validity of our dataset, we implemented a quality control procedure wherein any predicted sunshine duration value below 0 was reset to 0. This was necessary as the model's unconstrained regression output can generate physiologically impossible negative values near zero.

**Comment 9:**

The author mentioned the limitations of satellite remote sensing in estimating the meteorological variables in the "introduction". However, sunshine duration is greatly affected by cloud and aerosol parameters observed by remote sensing. When comparing sunshine duration products, please consider comparing sunshine duration datasets estimated based on remote sensing data (https://doi.org/10.5194/essd-17-1427-2025) and explain the advantages of the research method used in this study.

**Response:** Thank you very much for this insightful comment. Following the reviewer's suggestion, we have incorporated the Himawari AHI–based daily sunshine duration (SD) dataset (Zhang et al., 2025) into our comparative analysis. This satellite-derived, high-resolution product (5 km, 2016–2023) complements the homogenized station-based SSD dataset (2°, 1961–2022) and provides an independent benchmark for recent years.

To ensure logical consistency, Section 2.5 Existing gridded products for comparison in the Materials was rewritten to include the Himawari SD dataset alongside CMFD 2.0 and SSD, clearly outlining the rationale for selecting these complementary products. Furthermore, Section 4.3.2 Sunshine duration in the Results and Discussion was substantially revised to present a comprehensive comparison of our reconstructed dataset against both SSD and Himawari SD.

The revised analysis demonstrates that the reconstructed dataset achieves accuracy comparable to SSD in long-term temporal consistency, while also performing competitively with Himawari SD in recent high-resolution comparisons. Specifically, our reconstruction yields smaller systematic bias than Himawari, while Himawari attains slightly higher correlation in daily variability. These complementary findings highlight the robustness of the reconstruction framework and its combined strengths: reduced bias relative to satellite products, temporal stability comparable to homogenized long-term datasets, and the unique provision of six decades of 1 km daily sunshine duration fields for hydrometeorological applications in topographically complex regions.

The revised content of Section *2.5 Existing gridded products for comparison* is provided below:

"To assess the reliability and application potential of the reconstructed meteorological variables, representative and widely used gridded datasets were selected for comparison based on their scientific relevance and availability. Specifically, for average temperature, atmospheric pressure, and relative humidity, we employed the latest version of the China Meteorological Forcing Dataset (CMFD 2.0), whose earlier versions have been extensively used in land surface, hydrological, and ecological modeling over China (He et al., 2020).

The CMFD 2.0 (He et al., 2024) provides high-resolution (0.1°), 3-hourly gridded meteorological data for the period 1951–2020, covering the land area between 70°E–140°E and 15°N–55°N. It includes near-surface temperature, surface pressure, specific humidity, wind speed, radiation, and

precipitation. Compared to previous versions, CMFD 2.0 incorporates ERA5 reanalysis and station observations through updated data sources and artificial intelligence techniques, particularly for radiation and precipitation variables. It also introduces metadata on station relocations and expands the spatial coverage beyond China's borders, thereby improving temporal consistency and cross-regional applicability.

As CMFD 2.0 does not include sunshine duration, we incorporated two additional datasets for its evaluation. This step is critical because sunshine duration reconstruction constitutes the final step in our hierarchical framework, necessitating a thorough accuracy assessment to evaluate potential uncertainty propagation. To this end, we selected two complementary benchmarks: one long-term station-based product and one recent high-resolution satellite product. 1) The sunshine duration (SSD) dataset (He, 2024) serves as the long-term, station-based benchmark. It provides a homogenized daily sunshine duration record across China from 1961 to 2022 at a 2.0° × 2.0° resolution. Developed from over 2,200 meteorological stations and corrected for non-climatic influences (e.g., station relocations and instrumental changes), it offers a reliable baseline for evaluating the temporal stability and long-term climatological consistency of our reconstruction. 2) The Himawari AHI-based daily sunshine duration (SD) dataset (Zhang et al., 2025) provides a recent, high-resolution (5 km) satellite perspective for 2016–2023. It enables a direct assessment of our product's quality during the 2016–2019 overlap period and serves as a benchmark for evaluating fine-scale spatial accuracy."

The revised content of Section *4.3.2 Sunshine duration* is provided below:

"To comprehensively evaluate the accuracy of the reconstructed product, two representative benchmark datasets were employed: the homogenized station-based SSD product (2°) to assess long-term temporal consistency, and the high-resolution satellite-based Himawari SD product (5 km) to examine spatial performance.

As shown in Figure 8, when compared with the SSD dataset over 1961–2019, the reconstructed product demonstrated highly consistent accuracy. The median RMSE values were identical for both products (1.48 h), and the median CC values were likewise identical (0.93). The ME differed only slightly (0.05 h for the reconstructed dataset and 0.02 h for SSD), indicating comparable bias levels. Boxplot analysis further indicated that the reconstructed product exhibited slightly narrower interquartile ranges, whereas the SSD dataset showed fewer outliers in RMSE and CC. It should be

noted that although some of the 95 CMA validation stations may have been included in the SSD development, our reconstruction model excluded these stations from training, ensuring a higher degree of validation independence.

For spatial performance, the reconstructed dataset was compared with the Himawari SD dataset over the overlapping period of 2016–2019 (Figure 9). The evaluation was based on 91 stations, since three of the 95 validation stations had invalid sunshine duration values during this period and one station was located within the SD control region. Both products showed comparable RMSE levels (1.53 h for the reconstructed dataset compared with 1.48 h for Himawari). The satellite dataset achieved a slightly higher CC (0.94 compared with 0.92), reflecting stronger agreement in daily variations, while the reconstructed dataset exhibited a smaller ME (0.08 h compared with 0.21 h), indicating reduced bias.

[Figure]

**Figure 8: Boxplot comparison of RMSE, ME, and CC for sunshine duration between SSD (2.0°) and the reconstructed dataset developed in this study (1 km) from 1961 to 2019.**

[Figure]

**Figure 9: Boxplot comparison of RMSE, ME, and CC for sunshine duration between the Himawari AHI–based SD dataset (5 km) and the reconstructed dataset developed in this study (1 km) from 2016 to 2019.**

These complementary results indicate that the reconstruction framework can achieve accuracy

comparable to both a long-term homogenized station-based dataset and a high-resolution satellite-derived dataset."

**Comment 10:**

Is the model independent on a daily scale? Did the authors consider modeling based on different days of year (DOYs) to enhance the model's generalization ability in the future?

**Response:** We thank the reviewer for this helpful question. Our reconstruction framework is indeed independent on a daily scale: for each day, the model relies exclusively on station observations and spatial covariates corresponding to that specific day, without drawing on information from preceding or subsequent days. This design ensures that daily fields are generated without temporal autocorrelation, thereby simplifying interpretation and enhancing operational applicability. Moreover, because each day is reconstructed independently, occasional data gaps on specific days do not affect the performance on other days.

In fact, we note that we also tested an alternative transformer-based approach in which temporal context from surrounding days was incorporated. This experiment, however, showed limited skill in capturing day-to-day fluctuations compared to our daily-independent model. Because ESSD primarily emphasizes the quality and validation of datasets rather than extensive methodological comparisons, we did not include this exploratory test in the main text. We acknowledge that further exploration of temporal approaches, including the reviewer's suggestion to model based on different days of the year (DOY) to capture seasonal cycles, could be valuable for future improvements in long-term high-resolution meteorological reconstructions.

---

## Author Response (AR2)

**Dear Topic editor and reviewers,**

We sincerely thank the handling topic editor and the reviewers for their constructive comments and suggestions, which have helped us to significantly improve the clarity and rigor of our manuscript. Below, we provide a point-by-point response. The reviewers' comments are shown in **black**, and our responses are in **blue**. All changes have been incorporated and highlighted in **blue** in the marked-up manuscript.

**Responses to Reviewers' Comments:**

**Reviewer #1**

**Comment 1:**

This paper develops a long sequence dataset, and the research purpose and user target population of this dataset need to be further clarified;

**Response:** Thank you for your thoughtful comment emphasizing the need to explicitly state the research objectives and define the intended user community of the dataset. In response to this comment, we have revised the concluding paragraph of the Introduction (lines 81-91 of the revised manuscript) to provide a more explicit and direct statement of the dataset's research purpose and its target users. The revised text reads as follows:

"To address the limitations of existing meteorological datasets in spatial resolution, temporal continuity, and variable completeness, this study introduces a high-resolution dataset of daily near-surface meteorological variables — including average, maximum, and minimum air temperature, atmospheric pressure, relative humidity, and sunshine duration—across mainland China. Spanning six decades (1961–2021) with kilometer-level granularity, the dataset is designed to support fine-scale applications such as land surface modeling, drought assessment, and water resource management. It is particularly suited for both scientific investigations and operational decision-making in data-sparse and topographically complex regions, such as western China. To achieve this, a hierarchical and progressive reconstruction framework is implemented to generate gridded estimates of six variables at approximately 2 meters above ground level, based on in-situ observations and a 1 km digital elevation

model (DEM). A multilayer perceptron (MLP) regression model is employed in this framework to capture nonlinear relationships between station observations and topographic predictors (e.g., latitude, longitude, and elevation), enabling fine-scale reconstruction across complex terrain."

**Comment 2:**

"The objective of this study is to develop a high-resolution and accuracy-assessed dataset of daily near-surface meteorological variables across mainland China, suitable for applications in hydrological modeling, environmental monitoring, and climate analysis." How did the author consider the issue of temporal "homogeneity" in a long series dataset used for climate analysis?

**Response:** Thank you for raising this important point. We fully acknowledge that ensuring temporal homogeneity is essential for the reliability of long-term climate analyses. In this study, rather than applying direct spatial interpolation, we adopted a deep learning–based reconstruction framework that reconstructs each meteorological variable independently on a daily basis. The core of this framework is a multilayer perceptron (MLP) model, designed to learn and reconstruct the spatial distribution characteristics of each meteorological variable independently for every single day, based on nonlinear interactions with geographic and meteorological predictors. Because each day is modeled separately, potential quality issues on a particular day do not propagate temporally, thereby preserving the dataset's temporal integrity.

The in-situ observations used for training are sourced from the China Daily Surface Climate Dataset, developed and maintained by the China Meteorological Administration (CMA). According to the dataset documentation and metadata, this dataset has undergone comprehensive quality control and homogenization procedures to ensure its temporal consistency. Prior to model training, we further applied a quality-based filtering procedure to exclude all observations flagged with quality control codes indicating suspect (code 1), erroneous (code 2), missing (code 8), or unverified (code 9) values, thereby retaining only high-confidence records (code 0). This ensures the reliability of the training dataset and minimizes the propagation of observational uncertainties in the reconstruction process.

Owing to the day-by-day modeling strategy and the exclusive use of homogenized and quality-controlled station data, the resulting gridded product structurally maintains the temporal homogeneity

inherent in the original CMA dataset. This ensures that the dataset is well-suited for multi-decadal climate analyses and enables robust assessments of long-term climatic trends.

**Comment 3:**

What is the quality of the raw observation data used to establish 1km grid data? Has the author conducted data quality evaluation, analysis, quality control, homogenization processing, etc. on the original observation data during its use?

**Response:** Thank you for bringing up this important consideration. The accuracy and consistency of the raw observational data constitute the foundation for generating reliable gridded climate datasets, particularly for long-term applications. As noted in response to Comment 2, the reconstruction relies on in-situ records from the China Daily Surface Climate Dataset, developed and maintained by the CMA. According to the official dataset documentation and metadata, this dataset underwent extensive quality control and homogenization—particularly for the period 1951–2010, during which several nationwide campaigns were conducted to identify and correct erroneous or missing data and to ensure temporal consistency. Specifically, data from 1951 to 2010 were corrected and supplemented as part of a national data rescue initiative, involving repeated manual inspections, error correction, and recovery of missing records, resulting in a data availability rate exceeding 99% and near-perfect accuracy. From 2011 to mid-2012, a hierarchical three-level quality control system (station–provincial–national) was applied, while data after mid-2012 have undergone routine station-level validation.

In addition to leveraging this homogenized dataset, we applied a strict pre-filtering protocol prior to model training, excluding observations flagged with quality control codes indicating suspect (code 1), erroneous (code 2), missing or unmeasured (code 8), or unverified (code 9) values. Only high-confidence observations (code 0) were retained, thereby ensuring the integrity and robustness of the training dataset and minimizing the propagation of uncertainties in the final reconstruction.

Furthermore, recognizing that surface meteorological stations in China had experienced relocations and metadata updates over the years, we incorporated time-specific station coordinates—including dynamic longitude, latitude, and elevation values—for each daily observation. This design

avoids the spatial inaccuracies that could arise from using static station metadata and ensures that the model learns spatial relationships that faithfully represent the actual observational context on each day. This treatment helps maintain spatiotemporal consistency across the training samples and enhances the accuracy of fine-scale spatial reconstruction.

**Comment 4:**

Compared to a grid spatial resolution of 1km * 1km, using over 2000 observation data from China is relatively insufficient, especially in the sparse observation areas of western China. How does the author consider this issue?

**Response:** Thank you for your insightful comment. We fully acknowledge that the sparse distribution of meteorological stations—particularly across western China's complex terrain—poses a significant challenge for generating high-resolution (1-km) gridded meteorological products. Traditional interpolation methods, which often rely on linear assumptions and cannot fully incorporate topographic heterogeneity, tend to underperform in such data-scarce and topographically diverse regions.

To address this limitation, we adopted a model-driven reconstruction framework based on MLP, rather than relying on direct spatial interpolation. This framework is specifically designed to capture the nonlinear spatial distribution of each meteorological variable through a sequence of physically and statistically informed steps. By leveraging the point-wise spatial characteristics of each target variable, the model effectively learns fine-scale spatial structures across the domain. For example, air temperature is reconstructed solely from geographic predictors (latitude, longitude, elevation), while subsequent variables—such as atmospheric pressure, relative humidity, and sunshine duration—incorporate previously reconstructed variables as auxiliary inputs. This hierarchical structure enables the model to learn inter-variable dependencies and propagate spatial information from observation-rich regions to data-sparse areas.

Notably, in western China—characterized by rugged topography and limited station coverage—the model exhibits strong generalization capacity. Its ability to accurately reproduce fine-scale spatial patterns in these high-elevation regions provides empirical validation of the framework's robustness

under sparse observational constraints. Moreover, although no meteorological data from Taiwan were included during training, validation results in this region reveal high reconstruction accuracy, further underscoring the framework's ability to generalize learned spatial representations to previously unseen areas.

As part of our validation, we evaluated model performance across different regions using in-situ station data that were intentionally excluded from model training and reserved exclusively for validation purposes. The spatial distributions of key evaluation metrics (RMSE, ME, and CC) are presented in Figure 6 and discussed in detail in Lines 316–331 of the revised manuscript. These results indicate that, even in high-elevation and data-scarce regions such as the Tibetan Plateau and Taiwan—where no training stations were included—the reconstructed variables maintain reasonably high accuracy. This provides empirical support for the model's generalization capability and its applicability beyond the original training domain.

[Figure]

[Figure]

**Figure 6: Distribution maps of RMSE (a), ME (b) and CC (c) between grid-modelled data of six meteorological element products and in-situ data.**

**Comment 5:**

How is the " day boundary issues" handled? The ground meteorological observation in China adopts "20:00 Beijing time" as the boundary point of the day, which means that the observation day is from 20:00 to 20:00 the next day. This standard is applicable to daily value statistics of factors such as precipitation and temperature. Prior to the 1980s, some stations had a phenomenon of inconsistent day boundaries (such as a few stations using 08 or local time), which led to a decrease in comparability

between early data and other stations

**Response:** Thank you for your thoughtful comment regarding the potential inconsistency in day boundaries in Chinese meteorological data. We fully agree that variations in the definition of daily observation periods—if present—could affect the accuracy of reconstructed spatial patterns.

As stated in the documentation of the official daily meteorological dataset provided by the CMA, daily mean values are computed using observations recorded at 02:00, 08:00, 14:00, and 20:00 Beijing Time. Specifically, daily mean air temperature, relative humidity, and ground surface temperature are calculated as the average of these four values. For station pressure and wind speed, the same four-time averaging is generally applied; however, in the case of manual stations without automatic instruments, daily means are computed from three observations (08:00, 14:00, and 20:00). If any scheduled observation required for averaging is missing, the daily mean for that variable is flagged as missing. This standardized method ensures consistent temporal boundaries and comparability across stations.

Since the dataset documentation does not report any exceptions to this protocol, we consider the issue of day boundary consistency to be sufficiently addressed. Moreover, if the documentation had indicated that specific stations used non-standard day boundaries (e.g., using 08:00 or local time as the daily cutoff), we would have excluded those stations from our reconstruction and validation datasets to prevent potential biases. Therefore, the day boundary issue does not affect the reliability of our dataset.

**Comment 6:**

Overall evaluation: This long sequence dataset did not take into account the quality of the observation data used, day boundary issues, uniformity issues, etc. during the development process. Therefore, the dataset reconstructed in this article also has day boundary and uniformity issues, which will have a serious impact on downstream user research.

**Response:** Thank you very much for raising this important point. We fully understand the reviewer's concern regarding potential issues such as observation data quality, day-boundary inconsistencies, and temporal homogeneity in long-term meteorological datasets, as these aspects are indeed critical for ensuring the reliability of reconstructed products.

As elaborated in our responses to Comments 1 through 5, our reconstruction is based on the China

Surface Climate Daily Dataset, which is the national benchmark product released by the CMA. This dataset has undergone extensive quality control and homogenization procedures prior to public release. Specifically: (1) According to the official metadata, all daily values are calculated based on observations at 02:00, 08:00, 14:00, and 20:00 Beijing Time, ensuring standardized daily boundaries across the entire network; (2) The dataset documentation clearly states that data from 1951–2010 underwent repeated manual validation, correction, and gap-filling, resulting in >99% data availability and near-100% accuracy. For data after 2010, a standardized multi-level quality control procedure was applied, including station-, provincial-, and national-level checks, ensuring consistency and reliability across the full observation period. (3) For stations lacking full observations, daily means are flagged as missing, thereby preventing the inclusion of inconsistent data in our training or validation samples.

We believe these procedures reflect a robust national-level quality assurance protocol, which addresses many of the concerns raised regarding early inconsistencies and observation practices. We suggest that the issues raised in Comment 6—such as the quality of the observation data, day boundary definitions, and data homogeneity—have already been addressed in detail in our responses to Comments 1 through 5. In those responses, In those responses, we provided detailed explanations based on the official documentation and metadata descriptions of the CMA dataset, to clarify how such issues are handled in the source data. We sincerely hope those clarifications help to resolve any remaining concerns.

Finally, we fully appreciate the reviewer's attention to the integrity of long-term climate data. In the revised manuscript (Lines 97–99), we have added the following statement: "*According to the official documentation and metadata, these daily records are part of the CMA Surface Climate Daily Dataset, which follows a standardized observation protocol with unified day boundaries and homogenized records subjected to multi-tier quality control procedures.*" Therefore, we believe the reconstructed dataset does not suffer from unresolved day boundary or homogeneity issues and is reliable for downstream climate applications.

**Reviewer #2**

**General comment:**

This paper constructs a long-term meteorological variable dataset by decoding the nonlinear relationships between six meteorological variables and their spatial covariates. The method is innovative and the dataset is usable, but the paper needs to be revised based on the following points.

**Response:** We thank the reviewer for the positive overall assessment of our work, particularly the recognition of the methodological innovation and the usability of the dataset. We also appreciate the constructive suggestions for improvement, and we have carefully revised the manuscript to address all the specific points raised. Detailed responses are provided below.

**Comment 1:**

Abbreviations such as "CC" in the abstract should be spelled out in full.

**Response:** We acknowledge this helpful comment. In the revised manuscript, we have spelled out "CC" in full in the abstract as suggested. Moreover, we have also expanded other abbreviations (e.g., "RMSE" and "ME") at their first occurrence in accordance with academic writing standards. To avoid repetition, the revised abstract is provided in our response to Comment 2.

**Comment 2:**

The abstract does not adequately reflect the research objectives and significance of the study and needs to be improved.

**Response:** We appreciate this valuable comment. We agree that the abstract should explicitly highlight the research objectives and significance. In the revised version, we have emphasized these points at the beginning of the abstract. In addition, we have further refined the overall description to keep it concise and balanced, thereby improving its readability. The revised abstract is shown below:

"*The lack of high-accuracy, fine-resolution meteorological datasets in China has hindered progress in climate, hydrological, and ecological studies. In this study, we present a 1 km daily dataset spanning 1961–2021 across China, which includes six key variables—average, maximum, and*

*minimum temperature, atmospheric pressure, relative humidity, and sunshine duration—to provide a reliable foundation for advancing related research and applications. The dataset was generated using a novel hierarchical reconstruction framework that leveraged daily observations from 2345 meteorological stations and incorporated topographic attributes. This approach effectively decodes the nonlinear relationships between the meteorological variables and their spatial covariates, ensuring the generation of gridded daily fields that are both high-resolution and spatially continuous. Validation against 146 independent stations confirmed the high accuracy of the dataset. For average, maximum, and minimum temperatures, the errors are minimal (median root mean square errors (RMSEs): 1.16 ℃, 1.19 ℃, 1.29 ℃; median mean errors (MEs): -0.04 ℃, -0.10 ℃, -0.01 ℃), and the consistency with in-situ data is very high (median correlation coefficients (CCs): 0.99, 0.99, 0.99). Atmospheric pressure also shows very small errors (median RMSE: 2.65 hPa; median ME: -0.06 hPa) and strong correlation (median CC: 0.97). Relative humidity exhibits relatively lower accuracy (median RMSE: 6.33%; median ME: -0.52%; median CC: 0.90), but it still exceeds standard benchmarks. Sunshine duration maintains high precision (median RMSE: 1.48 h; median ME: 0.05 h; median CC: 0.93), indicating the robustness and reliability of the dataset. Further comparison reveals that in high-altitude and topographically complex regions, the reconstructed product demonstrates higher actual accuracy than suggested by station-to-grid validation, as spatial mismatches between stations and grid cells lead to systematic underestimation. Free access to the dataset is available at https://doi.org/10.11888/Atmos.tpdc.301341 or https://cstr.cn/18406.11.Atmos.tpdc.301341 (Zhao et al., 2024).*"

**Comment 3:**

The first paragraph of the introduction should provide some supporting citations.

**Response:** We thank the reviewer for this helpful suggestion. In the revised manuscript, we have added supporting citations in the first paragraph of the introduction to strengthen the background and provide authoritative references. Specifically, we now cite representative works showing how advances in computational power and remote sensing technologies have driven hydrological modeling toward more physically based and fully distributed simulations (Lettenmaier et al., 2015; Singh, 2018), climate change research across broader scales (IPCC, 2021), and studies emphasizing the importance

of high-resolution meteorological datasets in ungauged and topographically complex basins such as the Tibetan Plateau (Fu et al., 2020; Zhou et al., 2024). The revised paragraph is as follows:

"*With advances in computational power and remote sensing technologies, hydrological modeling has increasingly evolved toward fully distributed simulations (Lettenmaier et al., 2015; Singh, 2018), while climate change research continues to expand across broader spatial and temporal scales (IPCC, 2021). These developments have placed growing demands on the resolution and accuracy of basic meteorological inputs, particularly in ungauged and topographically complex basins such as the Tibetan Plateau (Fu et al., 2020; Zhou et al., 2024). High-resolution and high-quality meteorological datasets are essential for capturing fine-scale climate signals, representing land–atmosphere interactions, and supporting hydrological, ecological, and environmental assessments.*"

The newly added references are as follows:

1. Fu, Y., Ma, Y., Zhong, L., Yang, Y., Guo, X., Wang, C., Xu, X., Yang, K., Xu, X., Liu, L., Fan, G., Li, Y., and Wang, D.: Land-surface processes and summer-cloud-precipitation characteristics in the Tibetan Plateau and their effects on downstream weather: a review and perspective, National Science Review, 7, 500–515, https://doi.org/10.1093/nsr/nwz226, 2020.
2. IPCC: Climate Change 2021: The Physical Science Basis. Contribution of Working Group I to the Sixth Assessment Report of the Intergovernmental Panel on Climate Change, Masson-Delmotte, V., Zhai, P., Pirani, A., Connors, S. L., Péan, C., Berger, S., Caud, N., Chen, Y., Goldfarb, L., Gomis, M. I., Huang, M., Leitzell, K., Lonnoy, E., Matthews, J. B. R., Maycock, T. K., Waterfield, T., Yelekçi, O., Yu, R., and Zhou, B. (eds.), Cambridge University Press, Cambridge, United Kingdom and New York, NY, USA, 2391 pp., https://doi.org/10.1017/9781009157896, 2021.
3. Lettenmaier, D. P., Alsdorf, D., Dozier, J., Huffman, G. J., Pan, M., and Wood, E. F.: Inroads of remote sensing into hydrologic science during the WRR era, Water Resources Research, 51, 7309–7342, https://doi.org/10.1002/2015WR017616, 2015.
4. Singh, V. P.: Hydrologic modeling: progress and future directions, Geosci. Lett. 5, 15, https://doi.org/10.1186/s40562-018-0113-z, 2018.
5. Zhou, P., Tang, J., Ma, M., Ji, D., Shi, J.: High resolution Tibetan Plateau regional reanalysis 1961–present, Sci Data, 11, 444, https://doi.org/10.1038/s41597-024-03282-4, 2024.

**Comment 4:**

I noticed that the "Materials" section includes observation sites from different sources, but the verification sites were only selected from CMA. Have you considered selecting verification sites based on the weight of the number of sites from different sources?

**Response:** We sincerely thank the reviewer for raising this important point regarding the selection of validation sites. We appreciate the opportunity to clarify our methodology and believe the following

explanation adequately addresses the concern.

Our validation strategy was guided by two main objectives: (1) ensuring strict internal consistency in the training dataset and utilizing as many stations as possible for training (since deep learning algorithms require sufficient data to achieve good performance), and (2) achieving independence and robustness in validation. To this end, we employed 2,345 CMA stations for model training and withheld 95 independent CMA stations for validation. The selection of these 95 stations followed the principles described in Section 2.1, which state:

"*To support independent model validation, a total of 95 stations were selected as evaluation sites based on three principles: (1) ensuring geographical representativeness in terms of longitude, latitude, and elevation; (2) in densely monitored areas such as eastern China, a greater number of evaluation stations were retained without significantly reducing the size of the training dataset; and (3) in sparsely monitored regions such as western China (including Tibet and Xinjiang), the number of evaluation stations was intentionally reduced to ensure adequate data availability for model training.*"

However, we acknowledge that in key regions such as Taiwan and the Tibetan Plateau, CMA evaluation stations are very limited (with no stations in Taiwan). To address this, we supplemented the validation dataset with additional independent observations. Specifically, we incorporated 12 ground-based meteorological stations from the Department of Water Resources (DWR) located in the Tibetan Plateau region (see Section 2.2.1), and 8 international stations from the Global Surface Summary of Day (GSOD) dataset covering Taiwan (see Section 2.2.3). These supplementary data were added to ensure broader coverage and a more reliable evaluation in regions with sparse CMA observations.

In addition, for comparison against the China Meteorological Forcing Dataset (CMFD v2.0), we collected 31 field stations concentrated over the Tibetan Plateau from literature-based datasets archived at the National Tibetan Plateau Data Center (see Section 2.2.2). Since CMFD has assimilated or blended CMA data, and we cannot determine which specific CMA stations were used, these independent TP field stations were used as a fair and unbiased benchmark for inter-product comparison.

Through this combined strategy—using CMA-based stations for consistency and multi-source stations (DWR and GSOD) for robustness—we expanded the final validation set to 115 sites. In addition, 31 independent TP field stations from the National Tibetan Plateau Data Center were

specifically reserved for inter-product comparison to ensure fairness and independence. This approach provides a more comprehensive and balanced evaluation of our dataset, especially in high-altitude and data-sparse regions. The spatial distribution of both training and validation sites is shown in Figure 1.

We are grateful for the reviewer's comment, which allowed us to better articulate the rationale and implementation of our validation design.

**Comment 5:**

How to consider the cumulative error caused by progressively inputted meteorological variables during the modeling process, especially in the sunshine duration model.

**Response:** We thank the reviewer for raising this important point. We would like to clarify that cumulative error does not occur during the training stage, since each step of model training was performed using the true location information (longitude, latitude, and elevation) and observed meteorological values from CMA stations. For example, when training the mean temperature model, longitude, latitude, and elevation were used as predictors and the observed mean temperature as the target variable; when training the maximum temperature model, the observed mean temperature together with site attributes was used as input and the observed maximum temperature as the target. This procedure was applied similarly to other variables

But potential cumulative error could only arise during the reconstruction phase, when multiple 1 km gridded meteorological fields are generated from gridded mean temperature fields reconstructed using the 1 km DEM. Nevertheless, evaluation against independent ground observations demonstrated that the reconstructed products maintained high accuracy. In particular, sunshine duration — as the final step in the progressive framework, where cumulative error would theoretically be greatest — still exhibited consistently high precision (median RMSE: 1.48 h; median ME: 0.05 h; median CC: 0.93). This demonstrates the robustness and reliability of the dataset and indicates that, although error amplification is theoretically possible, it did not significantly affect the final results.

**Comment 6:**

How can authors reduce errors caused by the boundaries of the study area during modeling, given that

these areas have fewer observation stations?

**Response:** We gratefully acknowledge this important comment, which provides an opportunity to clarify our strategy for handling boundary effects. In our modeling process, we did not apply specific correction schemes to reduce potential boundary effects. However, the evaluation results demonstrate that the hierarchical deep-learning framework exhibits strong extrapolation and generalization ability. Taiwan provides a typical example of this capability: although no CMA stations were included in the training phase, the model was still able to reasonably reconstruct air temperature fields in this region. Independent validation using 8 international stations from the GSOD dataset (providing average, maximum, and minimum temperature) further confirmed the accuracy of the reconstruction.

We also analyzed this issue in the manuscript (see Section 4.2). As stated: "*The spatial distribution of RMSE, ME, and CC for all six meteorological variables are further illustrated in Figure 6, and consistent with expectations, the Subtropical and Southern Temperate Zones in southeastern China (STZ-southeastern China) display the best performance across all variables, largely due to the high density of training stations in these regions. In contrast, performance metrics are relatively lower in the Middle Temperate, Southern Temperate, and Plateau Climate Zones of northwestern China (MSPZ–northwest China), as well as in Taiwan, where no stations were included in training. Nevertheless, model performance in these regions remains robust. Notably, despite the absence of training data in Taiwan, the MLP model accurately reconstructs air temperature in that region, suggesting strong spatial generalizability.*"

**Comment 7:**

I noticed that Figure 4 contains a large amount of information, but the poor resolution and color quality make it difficult to see clearly. Please improve this.

**Response:** We appreciate this helpful suggestion. We have revised Figure 4 by increasing its resolution and optimizing the color scheme to enhance readability. The revised version is presented as follows:

[Figure]

**Figure 4: Line graphs of metrics (MSE, ME, CC) for optimal parameters in daily training and testing of MLP models from 1961 to 2021.**

**Comment 8:**

Please indicate whether adjustments were made when the author encountered situations where the sunshine duration was less than 0 during the verification.

**Response:** We gratefully acknowledge this thoughtful comment, which helped us clarify the handling of negative sunshine duration values. To ensure the physical validity of our dataset, we implemented a quality control step whereby any predicted sunshine duration value below 0 was reset to 0. This adjustment was necessary because the unconstrained regression output of the model can occasionally yield physically implausible negative values close to zero.

**Comment 9:**

The author mentioned the limitations of satellite remote sensing in estimating the meteorological variables in the "introduction". However, sunshine duration is greatly affected by cloud and aerosol parameters observed by remote sensing. When comparing sunshine duration products, please consider comparing sunshine duration datasets estimated based on remote sensing data (https://doi.org/10.5194/essd-17-1427-2025) and explain the advantages of the research method used in this study.

**Response:** Thank you very much for this insightful comment. Following the reviewer's suggestion, we have incorporated the Himawari AHI–based daily sunshine duration (SD) dataset (Zhang et al., 2025) into our comparative analysis. This satellite-derived, high-resolution product (5 km, 2016–2023) complements the homogenized station-based SSD dataset (2°, 1961–2022) and provides an independent benchmark for recent years.

To ensure logical consistency, Section 2.5 Existing gridded products for comparison in the Materials was rewritten to include the Himawari SD dataset alongside CMFD 2.0 and SSD, clearly outlining the rationale for selecting these complementary products. Furthermore, Section 4.3.2 Sunshine duration in the Results and Discussion was substantially revised to present a comprehensive comparison of our reconstructed dataset against both SSD and Himawari SD.

The revised analysis demonstrates that the reconstructed dataset achieves accuracy comparable to SSD in long-term temporal consistency, while also performing competitively with Himawari SD in

recent high-resolution comparisons. Specifically, our reconstruction yields smaller systematic bias than Himawari, while Himawari attains slightly higher correlation in daily variability. These complementary findings highlight the robustness of the reconstruction framework and its combined strengths: reduced bias relative to satellite products, temporal stability comparable to homogenized long-term datasets, and the unique provision of six decades of 1 km daily sunshine duration fields for hydrometeorological applications in topographically complex regions.

The revised content of Section *2.5 Existing gridded products for comparison* is provided below:

"*To assess the reliability and application potential of the reconstructed meteorological variables, representative and widely used gridded datasets were selected for comparison based on their scientific relevance and availability. Specifically, for average temperature, atmospheric pressure, and relative humidity, we employed the latest version of the China Meteorological Forcing Dataset (CMFD 2.0), whose earlier versions have been extensively used in land surface, hydrological, and ecological modeling over China (He et al., 2020).*

*The CMFD 2.0 (He et al., 2024) provides high-resolution (0.1°), 3-hourly gridded meteorological data for the period 1951–2020, covering the land area between 70°E–140°E and 15°N–55°N. It includes near-surface temperature, surface pressure, specific humidity, wind speed, radiation, and precipitation. Compared to previous versions, CMFD 2.0 incorporates ERA5 reanalysis and station observations through updated data sources and artificial intelligence techniques, particularly for radiation and precipitation variables. It also introduces metadata on station relocations and expands the spatial coverage beyond China's borders, thereby improving temporal consistency and cross-regional applicability.*

*As CMFD 2.0 does not include sunshine duration, we incorporated two additional datasets for its evaluation. This step is critical because sunshine duration reconstruction constitutes the final step in our hierarchical framework, necessitating a thorough accuracy assessment to evaluate potential uncertainty propagation. To this end, we selected two complementary benchmarks: one long-term station-based product and one recent high-resolution satellite product. 1) The sunshine duration (SSD) dataset (He, 2024) serves as the long-term, station-based benchmark. It provides a homogenized daily sunshine duration record across China from 1961 to 2022 at a 2.0° × 2.0° resolution. Developed from over 2,200 meteorological stations and corrected for non-climatic influences (e.g., station*

*relocations and instrumental changes), it offers a reliable baseline for evaluating the temporal stability and long-term climatological consistency of our reconstruction. 2) The Himawari AHI-based daily sunshine duration (SD) dataset (Zhang et al., 2025) provides a recent, high-resolution (5 km) satellite perspective for 2016–2023. It enables a direct assessment of our product's quality during the 2016–2019 overlap period and serves as a benchmark for evaluating fine-scale spatial accuracy.*"

The revised content of Section *4.3.2 Sunshine duration* is provided below:

"*To comprehensively evaluate the accuracy of the reconstructed product, two representative benchmark datasets were employed: the homogenized station-based SSD product (2°) to assess long-term temporal consistency, and the high-resolution satellite-based Himawari SD product (5 km) to examine spatial performance.*

*As shown in Figure 8, when compared with the SSD dataset over 1961–2019, the reconstructed product demonstrated highly consistent accuracy. The median RMSE values were identical for both products (1.48 h), and the median CC values were likewise identical (0.93). The ME differed only slightly (0.05 h for the reconstructed dataset and 0.02 h for SSD), indicating comparable bias levels. Boxplot analysis further indicated that the reconstructed product exhibited slightly narrower interquartile ranges, whereas the SSD dataset showed fewer outliers in RMSE and CC. It should be noted that although some of the 95 CMA validation stations may have been included in the SSD development, our reconstruction model excluded these stations from training, ensuring a higher degree of validation independence.*

*For spatial performance, the reconstructed dataset was compared with the Himawari SD dataset over the overlapping period of 2016–2019 (Figure 9). The evaluation was based on 91 stations, since three of the 95 validation stations had invalid sunshine duration values during this period and one station was located within the SD control region. Both products showed comparable RMSE levels (1.53 h for the reconstructed dataset compared with 1.48 h for Himawari). The satellite dataset achieved a slightly higher CC (0.94 compared with 0.92), reflecting stronger agreement in daily variations, while the reconstructed dataset exhibited a smaller ME (0.08 h compared with 0.21 h), indicating reduced bias.*

[Figure]

**Figure 8: Boxplot comparison of RMSE, ME, and CC for sunshine duration between SSD (2.0°) and the reconstructed dataset developed in this study (1 km) from 1961 to 2019.**

[Figure]

**Figure 9: Boxplot comparison of RMSE, ME, and CC for sunshine duration between the Himawari AHI–based SD dataset (5 km) and the reconstructed dataset developed in this study (1 km) from 2016 to 2019.**

*These complementary results indicate that the reconstruction framework can achieve accuracy comparable to both a long-term homogenized station-based dataset and a high-resolution satellite-derived dataset.*"

**Comment 10:**

Is the model independent on a daily scale? Did the authors consider modeling based on different days of year (DOYs) to enhance the model's generalization ability in the future?

**Response:** We thank the reviewer for this helpful question. Our reconstruction framework is independent at the daily scale: for each day, the model relies exclusively on station observations and spatial covariates corresponding to that specific day, without drawing on information from preceding or subsequent days. This design ensures that daily fields are generated without temporal autocorrelation, thereby simplifying interpretation and enhancing operational applicability. Moreover, because each day is reconstructed independently, occasional data gaps on specific days do not affect the performance on other days.

In fact, we also tested an alternative transformer-based approach in which temporal context from surrounding days was incorporated. This experiment, however, showed limited skill in capturing day-to-day fluctuations compared to our daily-independent model. Because ESSD primarily emphasizes the quality and validation of datasets rather than extensive methodological comparisons, we did not include this exploratory test in the main text. We acknowledge that further exploration of temporal approaches, including the reviewer's suggestion to model based on different days of the year (DOY) to capture seasonal cycles, could be valuable for future improvements in long-term high-resolution meteorological reconstructions.

**Reviewer #3**

**Comment 1:**

Regarding the selection of only 2 out of 12 field stations from the hourly land–atmosphere interaction dataset (Ma et al., 2024) for validation after quality control, it is unclear why the other stations were excluded. Please clarify the quality control criteria and explain whether the other stations were omitted due to poor data quality or other reasons.

**Response:** We greatly appreciate the reviewer's astute question, which provides us with a valuable opportunity to clarify our data processing workflow and underscore the robustness of our validation approach. The situation described stemmed from an initial technical oversight that was subsequently rectified through a comprehensive data collection effort. Please allow us to explain in detail.

(1) Initial Processing Oversight

We sincerely apologize for the lack of clarity in our original manuscript. During the initial data processing phase, an error in our automated script incorrectly led us to believe that only two stations from the Ma et al. (2024) dataset (NAMORS and Arou) had successfully passed our quality control (QC) procedures and were available for use. This was an unintentional technical mistake on our part.

(2) Proactive Expansion of Validation Data

To ensure the most robust validation possible, we were not satisfied with the limited number of stations initially retained. We therefore proactively sought out and incorporated every available source of ground observation data from the region. This extensive search enabled us to integrate additional

datasets, including 18 stations from the HiWATER network (Liu et al., 2018; Che et al., 2019) and 11 individual stations from other published studies (Zhang, 2018a,b; Gao, 2018; Luo, 2019; Ma, 2018; Wang and Wu, 2019; Luo and Zhu, 2020; Meng and Li, 2023). Through this expansion, our validation pool ultimately increased to a total of 31 station records from these combined sources.

(3) Discovery of Station Overlap and Final Validation Set

Upon integrating and cross-referencing all 31 records, we identified that five of the stations from our additional sources were duplicates of stations already contained within the Ma et al. (2024) dataset. In other words, a number of these stations represent the same physical locations that have been reported across different publications. For example, Arou, Yakou, Jingyangling, and Dashalong (from Ma et al., 2024) are also part of the HiWATER network; the QOMS station (cited from Ma, 2018) is included in the Ma et al. (2024) dataset; and the Maqu station (cited from Meng and Li, 2023) is likewise present in the Ma et al. (2024) dataset.

(4) Conclusion Regarding Data Quality and Selection

Therefore, to directly address the reviewer's question: the other stations from Ma et al. (2024) were not excluded due to poor data quality, but rather because of a technical error in our processing script. Some of these stations were later indirectly included through overlap with other published datasets (e.g., Arou, QOMS, Maqu). As a result, our final validation dataset comprises 31 stations from multiple independent sources, which we believe is sufficiently comprehensive to ensure the robustness and representativeness of the evaluation. We sincerely thank the reviewer again for prompting this important clarification.

We acknowledge that the original wording in Section 2.2.2 Literature-based datasets from the National Tibetan Plateau Data Center could be misleading. The phrase "(1) a publicly available dataset of hourly land–atmosphere interaction observations from 12 field stations (Ma et al., 2024), covering the period 2005–2021, from which 2 stations were selected after quality control for use as independent validation sites." may have unintentionally implied that the other 10 stations were excluded due to poor data quality, which was not the case. To avoid such ambiguity, we have revised the sentence to:

"*(1) a publicly available dataset of hourly land–atmosphere interaction observations (Ma et al.,*

*2024) , covering the period 2005–2021, of which two stations were employed as independent validation sites;*"

**Comment 2:**

The quality of the figures is suboptimal. Several figures lack units, and the x- and y-axis labels are missing or unclear. For instance, Figure 2 has low color contrast, making it difficult to distinguish between different elements. Improvements in figure clarity and completeness are necessary.

**Response:** We are grateful for this helpful observation. We fully agree that the clarity and completeness of the figures are critical for readers' understanding. In the revised manuscript, we have improved the figures by adding missing units, clarifying axis labels, and enhancing color contrast to make the elements more distinguishable. In addition, we have provided higher-resolution versions of all figures to further enhance their visual quality.

[Figure]

**Figure 1: The spatial distribution of training and evaluation meteorological stations in China.**

[Figure]

**Figure 2: The spatial distribution of training and evaluation meteorological stations in China.**

[Figure]

**Figure 4: Line graphs of metrics (MSE, ME, CC) for optimal parameters in daily training and testing of MLP models from 1961 to 2021.**

[Figure]

**Figure 5: Box plots of RMSE, ME, and CC for grid-modelled data of six meteorological element products and in-situ data.**

**(a) RMSE**

Average temperature — Unit: °C

Maximum temperature — Unit: °C

Minimum temperature — Unit: °C

● 0.49 - 1 ● 1 - 2 ● 2 - 3 ● 3 - 5 ● 5 - 7 ● 7 - 9.3

Atmospheric pressure — Unit: hpa

Relative humidity — Unit: %

Sunshine duration — Unit: h

● 0.8 - 4 ● 4 - 8 ● 8 - 15 ● 15 - 20 ● 20 - 41
● 3.6 - 5 ● 5 - 8 ● 8 - 12 ● 12 - 16 ● 16 - 31
● 1.2 - 1.5 ● 1.5 - 2 ● 2 - 3 ● 3 - 4.1

**(b) ME**

Average temperature — Unit: °C

Maximum temperature — Unit: °C

Minimum temperature — Unit: °C

● -6.8 - -4 ● -4 - -3 ● -3 - -2 ● -2 - -1 ● -1 - 0 ● 0 - 1 ● 1 - 2 ● 2 - 4

Atmospheric pressure — Unit: hpa

Relative humidity — Unit: %

Sunshine duration — Unit: h

● -41 - -20 ● -20 - -5 ● -5 - 0 ● 0 - 5 ● 5 - 15
● -29 - -10 ● -10 - -5 ● -5 - 0 ● 0 - 5 ● 5 - 11
● -0.4 - 0 ● 0 - 0.5 ● 0.5 - 1 ● 1 - 1.9

[Figure]

**Figure 6: Distribution maps of RMSE (a), ME (b) and CC (c) between grid-modelled data of six meteorological element products and in-situ data.**

[Figure]

**Figure 7: Boxplot comparison of RMSE, ME, and CC for average temperature, atmospheric pressure, and relative humidity between CMFD 2.0 and the reconstructed dataset developed in this study.**

[Figure]

**Figure 8: Boxplot comparison of RMSE, ME, and CC for sunshine duration between SSD (2.0°) and the reconstructed dataset developed in this study (1 km) from 1961 to 2019.**

[Figure]

**Figure 9: Boxplot comparison of RMSE, ME, and CC for sunshine duration between the Himawari AHI–based SD dataset (5 km) and the reconstructed dataset developed in this study (1 km) from 2016 to 2019.**

[Figure]

**Figure 10: Elevation differences between station elevations and corresponding DEM grid values: (a) spatial distribution, where red numbers denote station IDs with differences greater than 50 m; (b) point-line plot showing absolute elevation differences as a function of station ID.**

[Figure]

[Figure]

**Figure 11: Comparison of dotted line plots for RMSE, ME, and CC between in-situ data and station-based estimates, as well as between in-situ data and gridded data.**

[Figure]

**Figure 12: Annual spatial distribution of 6 meteorological elements in China from 1961 to 2019 based on daily reconstructed products.**

**Comment 3:**

The representativeness of the station data at the 1 km grid scale needs further discussion. Please elaborate on how the spatial representativeness of point stations affects the validation results, especially in regions with complex topography or sparse station coverage.

**Response:** We sincerely thank the reviewer for raising this insightful question, which directly addresses a core challenge in the validation of gridded products. We fully agree that the spatial representativeness of point stations is a fundamental factor—particularly in regions with complex terrain or sparse station coverage—and that it must be carefully considered when interpreting validation results.

This concern has also been central to our own research considerations. Drawing on our prior experience in evaluating satellite-based precipitation products, we consistently observed that mismatches between station locations and their corresponding grid cells—especially in terms of elevation—can introduce systematic biases into validation results. Motivated by this recognition, we

specifically designed Section 4.4 to investigate how such mismatches affect validation accuracy. In this section, we identified 36 stations located in high-relief regions where the elevation difference between recorded station elevations and those derived from the 1 km DEM exceeded 50 m, and conducted a controlled experiment in which two sets of predictions were generated: one using the actual station coordinates (longitude, latitude, and elevation) and the other using the coordinates of the corresponding grid-cell centers. By comparing both sets of predictions against in-situ measurements, we were able to explicitly separate and quantify the relative contributions of model error and representativeness error arising from elevation mismatch. The results demonstrated that for variables strongly influenced by elevation — such as temperature and pressure — representativeness error constitutes a substantial component of the total validation error, with its magnitude strongly correlated with the size of the elevation difference. These findings indicate that reduced validation accuracy in high-relief areas is not primarily due to deficiencies in the reconstruction framework itself, but rather to the inherent limitations of comparing point measurements with grid-cell estimates.

Despite these limitations, ground-based stations remain the cornerstone for validating gridded products—including satellite retrievals, reanalysis, and our reconstructions—as they provide the most accurate direct measurements available. In this context, the value of our work lies not in attempting to eliminate representativeness error, but in explicitly recognizing, quantifying, and interpreting it. Section 4.4 was designed with this purpose: to enable a fairer evaluation of model performance by distinguishing error sources attributable to environmental heterogeneity from those intrinsic to the reconstruction framework itself. Looking forward, we acknowledge that technological advances— such as denser ground-based networks and emerging mobile observation platforms (e.g., drones)— may help alleviate representativeness challenges.

**Comment 4:**

The reconstructed sunshine duration product does not show significant advantages over existing datasets. Concerns remain regarding data consistency, likely due to instrument changes and automation upgrades in CMA sunshine duration observations over time. This issue should be addressed to ensure reliability.

**Response:** Thank you very much for this insightful comment. In response to this concern, and consistent with another reviewer's constructive suggestion, we have incorporated the Himawari AHI–based daily sunshine duration (SD) dataset (Zhang et al., 2025) into our comparative analysis. This satellite-derived, high-resolution product (5 km, 2016–2023) complements the homogenized station-based SSD dataset (2°, 1961–2022) and provides an independent benchmark for recent years. The revised analysis demonstrates that the reconstructed dataset achieves accuracy comparable to SSD in long-term temporal consistency, while also performing competitively with Himawari SD in recent high-resolution comparisons. Specifically, our reconstruction yields smaller systematic bias than Himawari, while Himawari attains slightly higher correlation in daily variability. These complementary findings highlight the robustness of the reconstruction framework and its combined strengths: reduced bias relative to satellite products, temporal stability comparable to homogenized long-term datasets, and the unique provision of six decades of 1 km daily sunshine duration fields for hydrometeorological applications in topographically complex regions. The detailed revisions have been made in the following sections:

*Section 2.5 Existing gridded products for comparison:*

*"To assess the reliability and application potential of the reconstructed meteorological variables, representative and widely used gridded datasets were selected for comparison based on their scientific relevance and availability. Specifically, for average temperature, atmospheric pressure, and relative humidity, we employed the latest version of the China Meteorological Forcing Dataset (CMFD 2.0), whose earlier versions have been extensively used in land surface, hydrological, and ecological modeling over China (He et al., 2020).*

*The CMFD 2.0 (He et al., 2024) provides high-resolution (0.1°), 3-hourly gridded meteorological data for the period 1951–2020, covering the land area between 70°E–140°E and 15°N–55°N. It includes near-surface temperature, surface pressure, specific humidity, wind speed, radiation, and precipitation. Compared to previous versions, CMFD 2.0 incorporates ERA5 reanalysis and station observations through updated data sources and artificial intelligence techniques, particularly for radiation and precipitation variables. It also introduces metadata on station relocations and expands the spatial coverage beyond China's borders, thereby improving temporal consistency and cross-*

*regional applicability.*

*As CMFD 2.0 does not include sunshine duration, we incorporated two additional datasets for its evaluation. This step is critical because sunshine duration reconstruction constitutes the final step in our hierarchical framework, necessitating a thorough accuracy assessment to evaluate potential uncertainty propagation. To this end, we selected two complementary benchmarks: one long-term station-based product and one recent high-resolution satellite product. 1) The sunshine duration (SSD) dataset (He, 2024) serves as the long-term, station-based benchmark. It provides a homogenized daily sunshine duration record across China from 1961 to 2022 at a 2.0° × 2.0° resolution. Developed from over 2,200 meteorological stations and corrected for non-climatic influences (e.g., station relocations and instrumental changes), it offers a reliable baseline for evaluating the temporal stability and long-term climatological consistency of our reconstruction. 2) The Himawari AHI-based daily sunshine duration (SD) dataset (Zhang et al., 2025) provides a recent, high-resolution (5 km) satellite perspective for 2016–2023. It enables a direct assessment of our product's quality during the 2016–2019 overlap period and serves as a benchmark for evaluating fine-scale spatial accuracy.*"

*Section 4.3.2 Sunshine duration*:

"*To comprehensively evaluate the accuracy of the reconstructed product, two representative benchmark datasets were employed: the homogenized station-based SSD product (2°) to assess long-term temporal consistency, and the high-resolution satellite-based Himawari SD product (5 km) to examine spatial performance.*

*As shown in Figure 8, when compared with the SSD dataset over 1961–2019, the reconstructed product demonstrated highly consistent accuracy. The median RMSE values were identical for both products (1.48 h), and the median CC values were likewise identical (0.93). The ME differed only slightly (0.05 h for the reconstructed dataset and 0.02 h for SSD), indicating comparable bias levels. Boxplot analysis further indicated that the reconstructed product exhibited slightly narrower interquartile ranges, whereas the SSD dataset showed fewer outliers in RMSE and CC. It should be noted that although some of the 95 CMA validation stations may have been included in the SSD development, our reconstruction model excluded these stations from training, ensuring a higher degree of validation independence.*

*For spatial performance, the reconstructed dataset was compared with the Himawari SD dataset*

*over the overlapping period of 2016–2019 (Figure 9). The evaluation was based on 91 stations, since three of the 95 validation stations had invalid sunshine duration values during this period and one station was located within the SD control region. Both products showed comparable RMSE levels (1.53 h for the reconstructed dataset compared with 1.48 h for Himawari). The satellite dataset achieved a slightly higher CC (0.94 compared with 0.92), reflecting stronger agreement in daily variations, while the reconstructed dataset exhibited a smaller ME (0.08 h compared with 0.21 h), indicating reduced bias.*

[Figure]

**Figure 8: Boxplot comparison of RMSE, ME, and CC for sunshine duration between SSD (2.0°) and the reconstructed dataset developed in this study (1 km) from 1961 to 2019.**

[Figure]

**Figure 9: Boxplot comparison of RMSE, ME, and CC for sunshine duration between the Himawari AHI–based SD dataset (5 km) and the reconstructed dataset developed in this study (1 km) from 2016 to 2019.**

*These complementary results indicate that the reconstruction framework can achieve accuracy comparable to both a long-term homogenized station-based dataset and a high-resolution satellite-derived dataset.*"

In addition, we fully understand the reviewer's concern—instrument changes and automation upgrades are well-known factors affecting the homogeneity of long-term sunshine duration records. However, we would like to emphasize that the observational data used in this study are official CMA

station records, which have undergone standardized quality control and are widely recognized in the scientific community as the most reliable benchmark. Importantly, our reconstruction framework was explicitly designed to account for potential non-climatic biases arising from station relocations. Unlike conventional approaches, our model does not rely on fixed station metadata; instead, each daily observation in the training set is associated with the exact geographic information (longitude, latitude, and elevation) recorded for that specific date. As a result, if a station was relocated during 1961–2021, our model automatically treats the old and new locations as two separate geographic entities. This approach effectively avoids spurious biases introduced by site relocations and thereby improves the temporal consistency of the reconstructed product.

**Comment 5:**

The comparison with the CMFD product may be unfair, as CMFD utilizes a much smaller number of meteorological stations than this study. This discrepancy in input data may bias the comparison results. The authors should acknowledge this limitation and discuss its potential impact.

**Response:** Thank you very much for this valuable comment. We fully understand the reviewer's concern that differences in the number of input stations may affect the fairness of the comparison, and we agree that this point is indeed important.

We chose to compare our reconstruction with the CMFD product primarily because CMFD is widely used in China and has been recognized as an authoritative benchmark in the scientific community. Comparing a newly developed dataset against such a widely adopted reference is a common and necessary practice for evaluating its performance. Although CMFD is based on a relatively smaller number of stations, it achieves consistently high quality through the effective integration of reanalysis data and satellite products using advanced techniques, and has therefore become a well-recognized benchmark in this field.

We also fully acknowledge that machine learning methods depend on sufficiently large datasets, which is one of their inherent limitations. At the same time, a key advantage of such methods lies in their ability to capture the complex nonlinear relationships between meteorological variables and topographic or geographic factors. Therefore, in this study, the use of a larger set of ground-based observations is not only a necessary condition for applying the method, but also an important factor

that enables the reconstructed product to better capture spatial heterogeneity and to achieve accuracy comparable to, and in some respects even superior to, CMFD. These results highlight the potential of data-driven approaches to further improve gridded meteorological products. We will also mention this potential limitation in the discussion to ensure clarity for readers.

**Comment 6:**

The long-term trends and homogeneity of the reconstructed dataset are not discussed. An analysis of the temporal consistency and homogeneity of the data—especially concerning non-climatic factors such as station relocations or instrument changes—would strengthen the dataset's credibility.

**Response:** We sincerely thank the reviewer for this important comment. We fully understand and agree that the long-term trends and homogeneity of the dataset are critical for its credibility in climate-related applications.

As this is a data paper, our central objective is to provide and evaluate a high-quality, high-accuracy reconstructed dataset that is suitable for long-term climate analyses. To ensure its reliability, we used official station observations from the China Meteorological Administration as the basis. These data have undergone standardized quality control and are widely recognized within the scientific community. Importantly, our reconstruction framework was explicitly designed to account for potential non-climatic biases caused by station relocations. Unlike conventional approaches, our model does not rely on fixed station metadata; instead, each daily observation in the training set is associated with the exact geographic information (longitude, latitude, and elevation) recorded for that specific date. As a result, if a station was relocated during 1961–2021, the model automatically treated the old and new sites as two separate geographic entities, thereby minimizing spurious biases from relocations and improving temporal consistency.

For validation, we compared the reconstructed dataset against several widely used benchmark products, including CMFD 2.0, SSD, and Himawari SD. The results demonstrate that over a period of nearly 60 years, our dataset exhibits strong consistency with these benchmark products in terms of statistical measures (e.g., correlation coefficient CC, root-mean-square error RMSE) and long-term variability. Such cross-dataset consistency provides robust evidence of the temporal homogeneity and reliability of the reconstructed dataset, as further discussed in *Section 4.3.2 Sunshine duration*.

We once again thank the reviewer for the careful review and valuable suggestions, which provided us with the opportunity to clarify and better present this dataset.

**Other Revisions**

In our previous revision, following the handling topic editor's recommendation, we expanded the validation dataset by incorporating 29 additional independent field stations (see Section 2.2.2 *Literature-based datasets from the National Tibetan Plateau Data Center*). However, while these supplementary stations were described in the *Materials* and partly reflected in the comparative analyses against existing products, their results were not yet integrated into Section 4.2 *Validation of gridded meteorological element products using in-situ data* and Section 4.4 *Influence of elevation mismatch on validation accuracy*. To ensure consistency and completeness across the manuscript, we have incorporated the validation results of these additional stations into both sections in the current revised version. Corresponding updates were made to the related figures, statistical values, and numerical descriptions in the *Abstract* and Section *6 Conclusion*. These adjustments are minor and do not affect the overall evaluation accuracy of the dataset.

We would once again like to express our sincere gratitude to the editor for the careful coordination of the review process and the reviewers for generously devoting their time and expertise. We fully appreciate the considerable effort involved in reviewing and providing constructive feedback, which has been invaluable in improving both the rigor and clarity of our work. We have carefully addressed all comments and believe the manuscript has been materially improved as a result. Should any further clarification be required, we would be more than happy to provide it.